# Adversarial Schrödinger Bridge Matching

**Nikita Gushchin**[*]
Skoltech[†]
Moscow, Russia
n.gushchin@skoltech.ru

**Daniil Selikhanovych**[*]
Skoltech[†]
Moscow, Russia
selikhanovychdaniil@gmail.com

**Sergei Kholkin**[*]
Skoltech[†]
Moscow, Russia
s.kholkin@skoltech.ru

**Evgeny Burnaev**
Skoltech[†], AIRI [‡]
Moscow, Russia
e.burnaev@skoltech.ru

**Alexander Korotin**
Skoltech[†], AIRI[‡]
Moscow, Russia
a.korotin@skoltech.ru

## Abstract

The Schrödinger Bridge (SB) problem offers a powerful framework for combining optimal transport and diffusion models. A promising recent approach to solve the SB problem is the Iterative Markovian Fitting (IMF) procedure, which alternates between Markovian and reciprocal projections of continuous-time stochastic processes. However, the model built by the IMF procedure has a long inference time due to using many steps of numerical solvers for stochastic differential equations. To address this limitation, we propose a novel Discrete-time IMF (D-IMF) procedure in which learning of stochastic processes is replaced by learning just a few transition probabilities in discrete time. Its great advantage is that in practice it can be naturally implemented using the Denoising Diffusion GAN (DD-GAN), an already well-established adversarial generative modeling technique. We show that our D-IMF procedure can provide the same quality of unpaired domain translation as the IMF, using only several generation steps instead of hundreds. We provide the code at https://github.com/Daniil-Selikhanovych/ASBM.

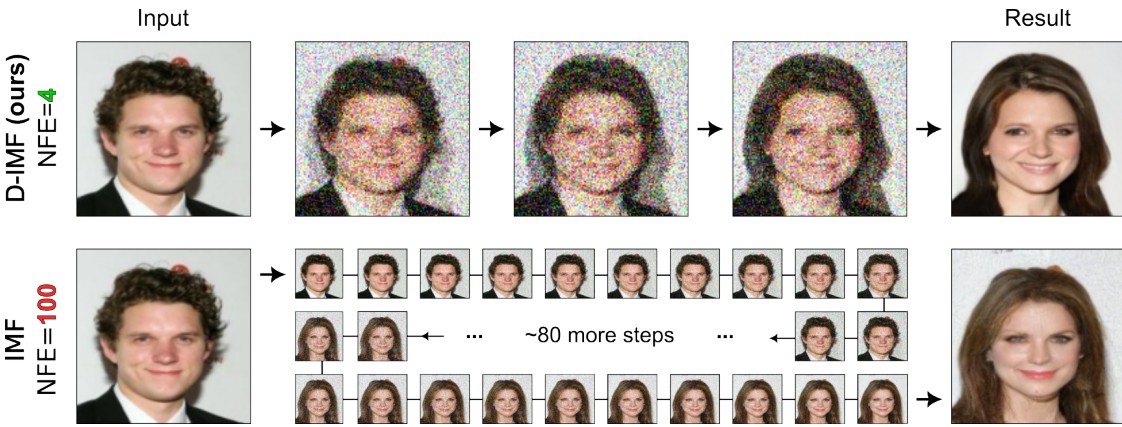

Figure 1: Our D-IMF approach performs unpaired image-to-image translation in just a few steps, achieving results comparable to the hundred-step IMF [47]. Celeba [33], *male→female* ($128 \times 128$).

---

[*]Equal contribution
[†]Skolkovo Institute of Science and Technology
[‡]Artificial Intelligence Research Institute

38th Conference on Neural Information Processing Systems (NeurIPS 2024).

# 1  Introduction

Recent generative models based on the Flow Matching [27] and Rectified Flows [30] show great potential as a successor of classical denoising diffusion models such as DDPM [15]. Both these approaches consider the same problem of learning an Ordinary Differential Equation (ODE) that interpolates one given distribution to the other one, e.g., noise to data. Thanks to the close connection to the theory of Optimal Transport (OT) problem [52], Flow Matching and Rectified Flows approaches typically have faster inference compared to classical diffusion models [32, 39]. Also, it was shown that they can outperform diffusion models on the high-resolution text-to-image synthesis: they even lie in the foundation of the recent Stable Diffusion 3 model [8].

The extension of Flow Matching and Rectified Flow approaches to the SDE are Bridge Matching (Markovian projection) and **Iterative Markovian fitting** (IMF) procedures [36, 47, 35], respectively. They also have a close connection with the OT theory. Specifically, it is known [47, 35] that IMF converges to the solution of the dynamic formulation of entropic optimal transport (EOT), also known as the Schrödinger Bridge (SB). However, learning continuous-time SDEs in IMF is non-trivial and, unfortunately, leads to **long inference** due to the necessity to use many steps of numerical solvers.

**Contributions.** This paper addresses the above-mentioned limitation of the existing Iterative Markovian Fitting (IMF) framework by introducing a novel approach to learn the Schrödinger Bridge.

1. **Theory I.** We introduce a Discrete Iterative Markovian Fitting (**D-IMF**) procedure (§3.2, 3.3), which innovatively applies discrete Markovian projection to solve the Schrödinger Bridge problem without relying on Stochastic Differential Equations. This approach significantly simplifies the inference process, enabling it to be accomplished (theoretically) in just a few evaluation steps.

2. **Theory II.** We derive closed-form update formulas for the D-IMF procedure when dealing with high-dimensional Gaussian distributions. This advancement permits a detailed empirical analysis of our method's convergence rate and enhances its theoretical foundation (§3.4, 4.1).

3. **Practice.** For general data distributions available by samples, we propose an algorithm (**ASBM**) to implement the discrete Markovian projection and our D-IMF procedure in practice (§4.2). Our algorithm is based on adversarial learning and Denoising Diffusion GAN [53]. Our learned SB model uses just 4 evaluation steps for inference (§3.5) instead of hundreds of the basic IMF [47].

**Notations.** In the paper, we simultaneously work with the continuous stochastic processes and discrete stochastic processes in the $D$-dimensional Euclidean space $\mathbb{R}^D$. We denote by $\mathcal{P}(C([0,1]),\mathbb{R}^D)$ the set of continuous stochastic processes with time $t \in [0,1]$, i.e., the set of distributions on continuous trajectories $f : [0,1] \to \mathbb{R}^D$. We use $dW_t$ to denote the differential of the standard Wiener process.

To establish a link between continuous and discrete stochastic processes, we fix $N \geq 1$ intermediate time moments $0 = t_0 < t_1 < \cdots < t_N < t_{N+1} = 1$ together with $t_0 = 0$ and $t_{N+1} = 1$. We consider discrete stochastic processes with those time-moments as the elements of the set $\mathcal{P}(\mathbb{R}^{D \times (N+2)})$ of probability distributions on $\mathbb{R}^{D \times (N+2)}$. Among such discrete processes, we are specifically interested in subset $\mathcal{P}_{2,ac}(\mathbb{R}^{D \times (N+2)}) \subset \mathcal{P}(\mathbb{R}^{D \times (N+2)})$ of absolutely continuous distributions on $\mathbb{R}^{D \times (N+2)}$ which have a finite second moment and entropy. For any such $q \in \mathcal{P}_{2,ac}(\mathbb{R}^{D \times (N+2)})$, we write $q(x_0, x_{t_1}, \ldots, x_{t_{N+1}})$ to denote its density at a point $(x_0, x_{t_1}, \ldots, x_{t_N}, x_1) \in \mathbb{R}^{D \times (N+2)}$. For continuous process $T$, we denote by $p^T \in \mathcal{P}(\mathbb{R}^{D \times (N+2)})$ the discrete process which is the finite-dimensional projection of $T$ to time moments $0 = t_0 < t_1 < \cdots < t_N < t_{N+1} = 1$. For convenience we also use the notation $x_{\text{in}} = (x_{t_1}, \ldots, x_{t_N})$ to denote the vector of all intermediate-time variables. In what follows, KL is a short notation for the Kullback-Leibler divergence.

# 2  Background

We start with recalling the Bridge Matching and Iterative Propotional Fitting procedures developed for continuous-time stochastic processes (§2.1). Next, we discuss the Schrödinger Bridge problem, the solution to which is the unique fixed point of Iterative Markovian Fitting procedure (§2.2).

## 2.1  Bridge Matching and Iterative Markovian Fitting Procedures

Modern diffusion and flow generative modeling are mainly about the construction of a model that interpolates one probability distribution $p_0 \in \mathcal{P}_{2,ac}(\mathbb{R}^D)$ to some another probability distribution $p_1 \in \mathcal{P}_{2,ac}(\mathbb{R}^D)$. One of the general approaches for this task is the Bridge Matching [29, 31, 3].

**Reciprocal processes.** The Bridge Matching procedure is applied to the processes, which are represented as a mixture of Brownian Bridges. Consider the Wiener process $W^\epsilon$ with the volatility $\epsilon$ which start at $p_0$, i.e., the process given by the SDE: $dx_t = \sqrt{\epsilon} dW_t$, $x_0 \sim p_0$. Let $W^\epsilon_{|x_0,x_1}$ denote

the stochastic process $W^\epsilon$ conditioned on values $x_0, x_1$ at times $t = 0, 1$, respectively. This process $W^\epsilon_{|x_0, x}$ is called the Brownian Bridge [17, Chapter 9]. For some $q(x_0, x_1) \in \mathcal{P}_{2,ac}(\mathbb{R}^{D \times 2})$ with $q(x_0) = p_0(x_0)$ and $q(x_1) = p_1(x_1)$ the process $T_q \stackrel{\text{def}}{=} \int W^\epsilon_{|x_0, x_1} dq(x_0, x_1)$ is called the mixture of Brownian Bridges. Following [47], we say that mixtures of Brownian Bridges form a *reciprocal class* of processes (for the Brownian Bridge). For brevity, we call these processes just reciprocal processes.

**Bridge matching [29, 31].** The goal of Bridge Matching (with the Brownian Bridge) is to construct continuous-time Markovian process $M$ from $p_0$ to $p_1$ in the form of SDE: $dx_t = v(x_t, t)dt + \sqrt{\epsilon}dW_t$. This is achieved by using the *Markovian projection* of a reciprocal process $T_q = \int W^\epsilon_{|x_0, x_1} dq(x_0, x_1)$, which aims to find the Markovian process $M$ which is the most similar to $T_q$ in the sense of KL:

$$\text{proj}_\mathcal{M}(T_q) \stackrel{\text{def}}{=} \underset{M \in \mathcal{M}}{\arg\min} \text{KL}\left(T_q \| M\right),$$

where $\mathcal{M} \subset \mathcal{P}(C([0, 1]), \mathbb{R}^D)$ is the set of all Markovian processes. For the Brownian Bridge $W^\epsilon_{|x_0, x_1}$ it is known [47, 11] that the SDE and the drift $v(x_t, t)$ of $\text{proj}_\mathcal{M}(T_q)$ is given by:

$$dx_t = v(x_t, t)dt + \sqrt{\epsilon}dW_t, \quad v(x_t, t) = \int \frac{x_1 - x_t}{1 - t} p^{T_q}(x_1 | x_t)dx_1,$$

where $p^{T_q}(x_1 | x_t)$ the conditional distribution of the stochastic process $T_q$ at time moments $t$ and 1. The process $\text{proj}_\mathcal{M}(T_q)$ has the same time marginal distributions $p^{T_q}(x_t)$ as the original Brownian bridge mixture $T_q$. However, the joint distribution $p^{T_q}(x_0, x_1)$ of $T_q$ and the joint distribution $p^{\text{proj}_\mathcal{M}(T_q)}(x_0, x_1)$ of its projection $\text{proj}_\mathcal{M}(T_q)$ do not coincide in the general case [6], see Figure 2.

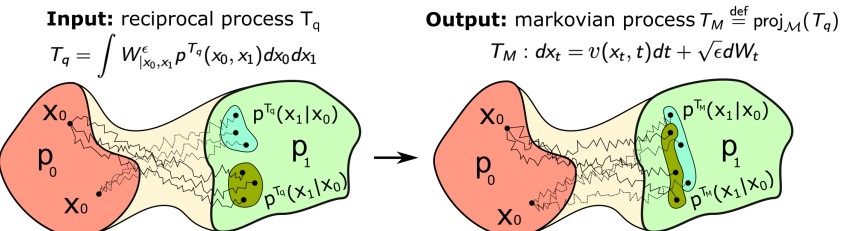

Figure 2: Markovian projection of a reciprocal stochastic process $T_q$.

**Iterative Markovian Fitting [47, 35, 1].** The Iterative Markovian Fitting procedure introduces a second type of projection of continuous-time stochastic processes called the *Reciprocal projection*. For a process $T$, it is is defined by $\text{proj}_\mathcal{R}(T) = \int W^\epsilon_{|x_0, x_1} dp^T(x_0, x_1)$, see illustrative Figure 3.

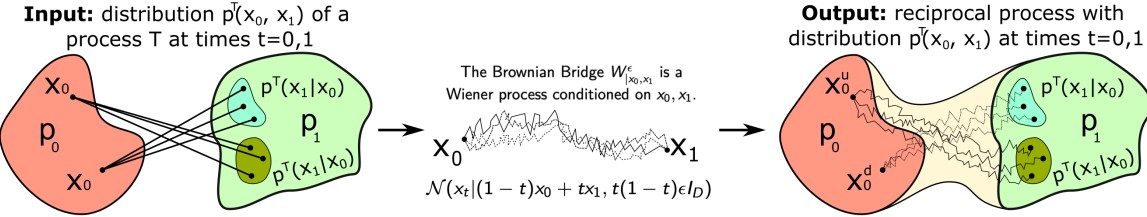

Figure 3: Reciprocal projection of a stochastic process $T$, i.e., $\text{proj}_\mathcal{R}(T) = \int W^\epsilon_{|x_0, x_1} dp^T(x_0, x_1)$.

The process $\text{proj}_\mathcal{R}(T)$ is called a projection, since:

$$\text{proj}_\mathcal{R}(T) = \underset{R \in \mathcal{R}}{\arg\min} \text{KL}\left(T \| R\right),$$

where $\mathcal{R} \subset \mathcal{P}(C([0, 1]), \mathbb{R}^D)$ is the set of all reciprocal processes. The Iterative Markovian Fitting procedure is an alternation between Markovian and Reciprocal projections:

$$T^{2l+1} = \text{proj}_\mathcal{M}(T^{2l}), \quad T^{2l+2} = \text{proj}_\mathcal{R}(T^{2l+1}),$$

It is known that the procedure converges to the unique stochastic process $T^*$, which is known as a solution to the Schrödinger Bridge (SB) problem between $p_0$ and $p_1$. Furthermore, the SB $T^*$ is the only process starting at $p_0$ and ending at $p_1$ that is both Markovian and reciprocal [25].

## 2.2 Schrödinger Bridge (SB) Problem

**Schrödinger Bridge problem.** The Schrödinger Bridge problem [44] was proposed in 1931/1932 by Erwin Schrödinger. For the Wiener prior $W^\epsilon$ Schrödinger Bridge problem between two probability distributions $p_0 \in \mathcal{P}_{2,ac}(\mathbb{R}^D)$ and $p_1 \in \mathcal{P}_{2,ac}(\mathbb{R}^D)$ is to minimize the following objective:

$$\min_{T \in \mathcal{F}(p_0,p_1)} \text{KL}\left(T \| W^\epsilon\right), \tag{1}$$

where $\mathcal{F}(p_0,p_1) \subset \mathcal{P}(C([0,1]), \mathbb{R}^D)$ is the subset of stochastic processes which starts at distribution $p_0$ (at the time $t = 0$) and end at $p_1$ (at $t = 1$). The Scrhödinger Bridge has a unique solution, which is a diffusion process $T^*$ described by the SDE: $dX_t = v^*(X_t, t)dt + \sqrt{\epsilon}dW_t$ [25]. The process $T^*$ is called *the Schrödinger Bridge* and $v^* : \mathbb{R}^D \times [0,1] \to \mathbb{R}^D$ is called *the optimal drift*.

From the practical point of view, the solution to the SB problem $T^*$ tends to preserve the Euclidean distance between start point $x_0$ and endpoint $x_1$. The equivalent form of SB problem, the static Schrödinger Bridge problem, explains this property more clearly.

**Static Schrödinger Bridge problem.** One may decompose $\text{KL}(T\|W^\epsilon)$ as [51, Appendix C]:

$$\text{KL}(T\|W^\epsilon) = \text{KL}\left(p^T(x_0,x_1)\|p^{W^\epsilon}(x_0,x_1)\right) + \int \text{KL}(T_{|x_0,x_1}\|W^\epsilon_{|x_0,x_1})dp^T(x_0,x_1), \tag{2}$$

i.e., KL divergence between $T$ and $W^\epsilon$ is a sum of two terms: the 1st represents the similarity of the processes' joint marginal distributions at start and finish times $t = 0, 1$, while the 2nd term represents the average similarity of conditional processes $T_{|x_0,x_1}$ and $W^\epsilon_{|x_0,x_1}$. In [25, Proposition 2.3], the authors show that if $T^*$ solves (1), then $T^*_{|x_0,x_1} = W^\epsilon_{|x_0,x_1}$. Hence, one may optimize (1) over $T$ for which $T_{|x_0,x_1} = W^\epsilon_{|x_0,x_1}$ for every $x_0, x_1$, i.e., over reciprocal processes $T$:

$$(1) = \min_{T \in \mathcal{F}(p_0,p_1) \cap \mathcal{R}} \text{KL}\left(p^T(x_0,x_1)\|p^{W^\epsilon}(x_0,x_1)\right) = \min_{q \in \Pi(p_0,p_1)} \text{KL}\left(q(x_0,x_1)\|p^{W^\epsilon}(x_0,x_1)\right), \tag{3}$$

where $\Pi(p_0,p_1) \subset \mathcal{P}_{2,ac}(\mathbb{R}^{D \times 2})$ is the set of joint probability distributions with marginal distributions $p_0$ and $p_1$. Thus, the initial Schrödinger Bridge problem can be solved by optimizing only over a reciprocal process's joint distribution $q(x_0,x_1)$ at $t = 0, 1$. This problem is called the Static Schrödinger Bridge problem. In turn, the problem can be rewritten in the following way [12, Eq. 7]:

$$\min_{q \in \Pi(p_0,p_1)} \epsilon \text{KL}(q\|p^{W^\epsilon}(x_0,x_1)) = \min_{q \in \Pi(p_0,p_1)} \int \frac{||x_0 - x_1||^2}{2} dq(x_0,x_1) - \epsilon \cdot \text{Entropy}(q) + C, \tag{4}$$

i.e., as finding a joint distribution $q(x_0,x_1)$ which tries to minimize the Euclidian distance $\frac{||x-y||^2}{2}$ between $x_0$ and $x_1$ (preserve similarity between $x_0$ and $x_1$), but with the addition of entropy regularizer $\epsilon \cdot \text{Entropy}(q)$ with the coefficient $\epsilon$. Thus, the coefficient $\epsilon > 0$, which is the same for all problems considered above, regulates the stochastic or diversity of samples from $q(x_0,x_1)$. The last problem (4) is also known as the entropic optimal transport (EOT) problem [4, 38, 25].

# 3 Adversarial Schrödinger Bridge Matching (ASBM)

The IMF framework [35, 47] works with *continuous* time stochastic processes: it is built on the well-celebrated result that the only process which is both Markovian and reciprocal is the Schrödinger bridge $T^*$ [25]. We derive an analogous theoretical result but for processes in *discrete* time. We provide proofs for all the theorems and propositions in Appendix B.

In §3.1, we give preliminaries on discrete processes with Markovian and reciprocal properties. In §3.2, we present the main theorem of our paper, which is the foundation of our **Discrete-time Iteratime Markovian Fitting (D-IMF)** framework. In §3.3, we describe D-IMF procedure itself and prove that it allows us to solve the Schrödinger Bridge problem. In §3.4, we provide an analysis of applying our D-IMF for solving the Schrödinger Bridge between Gaussian distributions. In §3.5, we present the practical implementation of our D-IMF procedure using adversarial learning.

## 3.1 Discrete Markovian and reciprocal stochastic processes

**Discrete reciprocal processes.** We define the discrete reciprocal processes similarly to the continuous case by considering the finite-time projection of the Brownian bridge $W^\epsilon_{|x_0,x_1}$, which is given by:

$$p^{W^\epsilon}(x_{t_1}, \ldots, x_{t_N} | x_0, x_1) = \prod_{n=1}^{N} p^{W^\epsilon}(x_{t_n} | x_{t_{n-1}}, x_1), \tag{5}$$

$$p^{W^\epsilon}(x_{t_n}|x_{t_{n-1}}, x_1) = \mathcal{N}(x_{t_n}|x_{t_{n-1}} + \frac{t_n - t_{n-1}}{1 - t_{n-1}}(x_1 - x_{t_{n-1}}), \epsilon \frac{(t_n - t_{n-1})(1 - t_n)}{1 - t_{n-1}}). \qquad (6)$$

This joint distribution $p^{W^\epsilon}(x_{t_1}, \ldots, x_{t_N}|x_0, x_1)$ defines a discrete stochastic process, which we call a discrete Brownian bridge. In turn, we say that a distribution $q \in \mathcal{P}_{2,ac}(\mathbb{R}^{D \times (N \times 2)})$ is a mixture of discrete Brownian bridges if it satisfies

$$q(x_0, x_{t_1}, \ldots, x_{t_N}, x_1) = p^{W^\epsilon}(x_{t_1}, \ldots, x_{t_N}|x_0, x_1)q(x_0, x_1),$$

where $q(x_0, x_1)$ denotes its joint marginal distribution of $q$ at times $0, 1$. That is, its "inner" part at times $t_1, \ldots, t_N$ is the discrete Brownian Bridge. We denote the set of all such mixtures as $\mathcal{R}(N) \subset \mathcal{P}_{2,ac}(\mathbb{R}^{D \times (N+2)})$ and call them discrete reciprocal processes.

**Discrete Markovian processes.** We say that a discrete process $q \in \mathcal{P}_{2,ac}(\mathbb{R}^{D \times (N+2)})$ is Markovian if its density can be represented in the following form (recall that $t_0 = 0, t_{N+1} = 1$):

$$q(x_0, x_{t_1}, x_{t_2}, \ldots, x_{t_N}, x_1) = q(x_0) \prod_{n=1}^{N+1} q(x_{t_n}|x_{t_{n-1}}). \qquad (7)$$

We denote the set of all such discrete Markovian processes as $\mathcal{M}(N) \subset \mathcal{P}_{2,ac}(\mathbb{R}^{D \times (N+2)})$.

## 3.2 Main Theorem

**Theorem 3.1** (Discrete Markovian and reciprocal process is the solution of static SB). *Consider any discrete process $q \in \mathcal{P}_{2,ac}(\mathbb{R}^{D \times (N+2)})$, which is simultaneously reciprocal and markovian, i.e. $q \in \mathcal{R}(N)$ and $q \in \mathcal{M}(N)$ and has marginals $q(x_0) = p_0(x_0)$ and $q(x_1) = p_1(x_1)$:*

$$q(x_0, x_{t_1}, \ldots, x_{t_N}, x_1) = p^{W^\epsilon}(x_{t_1}, \ldots, x_{t_N}|x_0, x_1)q(x_0, x_1) = q(x_0) \prod_{n=1}^{N+1} q(x_{t_n}|x_{t_{n-1}}),$$

*Then $q(x_0, x_{t_1}, \ldots, x_{t_N}, x_1) = p^{T^*}(x_0, x_{t_1}, \ldots, x_{t_N}, x_1)$, i.e., it is the finite-dimensional projection of the Schrödinger Bridge $T^*$ to the considered times. Moreover, its joint marginal $q(x_0, x_1)$ at times $t = 0, 1$ is the solution to the **static SB** problem* (4) *between $p_0$ and $p_1$, i.e., $q(x_0, x_1) = p^{T^*}(x_0, x_1)$.*

Thus, to solve the static SB problem, it is enough to find a Markovian mixture of discrete Brownian bridges. To do so, we propose the Discrete-time Iterative Markovian Fitting (D-IMF) procedure.

## 3.3 Discrete-time Iterative Markovian Fitting (D-IMF) procedure

Similar to the IMF procedure, our proposed Discrete-time IMF is based on two alternating projections of discrete stochastic processes: reciprocal and Markovian. We start with the reciprocal projection.

**Definition 3.2** (Discrete Reciprocal Projection). Assume that $q \in \mathcal{P}_{2,ac}(\mathbb{R}^{D \times (N+2)})$ is a discrete stochastic process. Then the reciprocal projection $\text{proj}_{\mathcal{R}}(q)$ is a discrete stochastic process with the joint distribution given by:

$$\left[\text{proj}_{\mathcal{R}}(q)\right](x_0, x_{t_1}, \ldots, x_{t_N}, x_1) = p^{W^\epsilon}(x_{t_1}, \ldots, x_{t_N}|x_0, x_1)q(x_0, x_1). \qquad (8)$$

This projection takes the joint distribution of start and end points $q(x_0, x_1)$ and inserts the Brownian Bridge for intermediate time moments, see Figure 4. The prop. below justifies the projection's name.

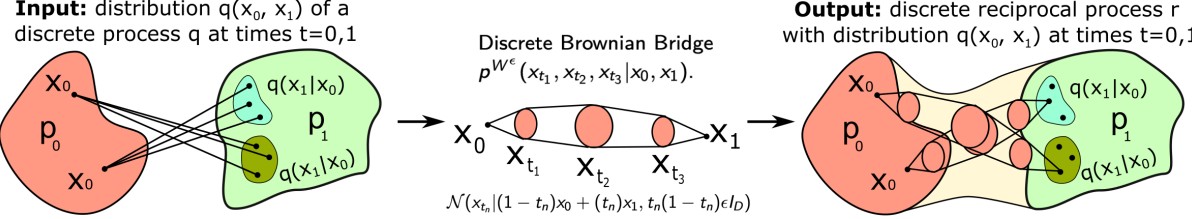

Figure 4: Reciprocal projection of a discrete stochastic process $q$, i.e.,
$$r(x_0, x_{t_1}, \ldots, x_{t_N}, x_1) = p^{W^\epsilon}(x_{t_1}, \ldots, x_{t_N}|x_0, x_1)q(x_0, x_1).$$

**Proposition 3.3** (Discrete Reciprocal projection minimizes KL divergence with reciprocal processes). *Under mild assumptions, the reciprocal projection $proj_{\mathcal{R}}(q)$ of a stochastic discrete process $q \in \mathcal{P}_{2,ac}(\mathbb{R}^{D\times(N+2)})$ is the unique solution for the following optimization problem:*

$$proj_{\mathcal{R}}(q) = \underset{r \in \mathcal{R}(N)}{\arg\min}\, KL\left(q\|r\right). \tag{9}$$

Similarly to the discrete reciprocal projection, we introduce discrete Markovian projection.

**Definition 3.4** (Discrete Markovian Projection). Assume that $q \in \mathcal{P}_{2,ac}(\mathbb{R}^{D\times(N+2)})$ is a discrete stochastic process. The Markovian projection of $q$ is a discrete stochastic process $proj_{\mathcal{M}}(q) \in \mathcal{P}_{2,ac}(\mathbb{R}^{D\times(N+2)})$ whose joint distribution given by:

$$\left[proj_{\mathcal{M}}(q)\right](x_0, x_{t_1}, ..., x_{t_N}, x_1) = q(x_0)\prod_{n=1}^{N+1} q(x_{t_n}|x_{t_{n-1}}). \tag{10}$$

Despite it is possible to use any discrete stochastic process $q$ as an input to a discrete markovian projection, in the rest of the paper only discrete reciprocal processes are considered as an input. For such cases, we provide a visualization of the markovian projection in Figure 5.

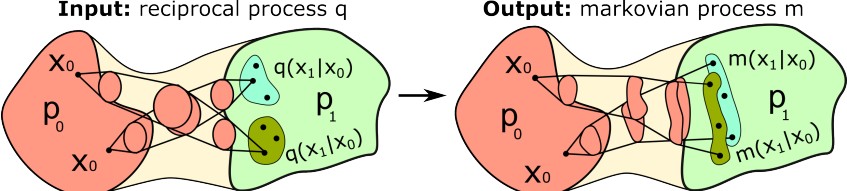

Figure 5: Markovian projection of a reciprocal discrete stochastic process $q$.

As with the reciprocal projection, our following proposition justifies the name of the projection.

**Proposition 3.5** (Discrete Markovian projection minimizes KL divergence with Markovian processes). *Under mild assumptions, the Markovian projection $proj_{\mathcal{M}}(q)$ of a stochastic discrete process $q \in \mathcal{P}_{2,ac}(\mathbb{R}^{D\times(N+2)})$ is a unique solution to the following optimization problem:*

$$proj_{\mathcal{M}}(q) = \underset{m \in \mathcal{M}(N)}{\arg\min}\, KL\left(q\|m\right). \tag{11}$$

Now we are ready to define our D-IMF procedure. For two given distributions $p_0 \in \mathcal{P}_{2,ac}(\mathbb{R}^D)$ and $p_1 \in \mathcal{P}_{2,ac}(\mathbb{R}^D)$ at times $t = 0$ and $t = 1$, respectively, it starts with any discrete Brownian mixture $p^{W^\epsilon}(x_{t_1}, \ldots, x_{t_N}|x_0, x_1)q(x_0, x_1)$, where $q(x_0, x_1) \in \Pi(p_0, p_1) \cap \mathcal{P}_{2,ac}(\mathbb{R}^{D\times 2})$. Then, it constructs the following sequence of discrete stochastic processes:

$$q^{2l+1} = proj_{\mathcal{M}}(q^{2l}), \quad q^{2l+2} = proj_{\mathcal{R}}(q^{2l+1}). \tag{12}$$

**Theorem 3.6** (D-IMF procedure converges to the the Schrödinger Bridge). *Under mild assumptions, the sequence $q^l$ constructed by our D-IMF procedure converges in KL to $p^{T^*}$. In particular, $q^l(x_0, x_1)$ convergence to the solution $p^{T^*}(x_0, x_1)$ of the static SB. Namely, we have*

$$\lim_{l\to\infty} KL\left(q^l\|p^{T^*}\right) = 0, \qquad \text{and} \qquad \lim_{l\to\infty} KL\left(q^l(x_0, x_1)\|p^{T^*}(x_0, x_1)\right) = 0.$$

### 3.4 Closed form Updates of D-IMF for Gaussian Distributions

In this section, we show that our D-IMF updates (12) can be derived in the closed form for the Gaussian case. Let $p_0 = \mathcal{N}(x_0|\mu_0, \Sigma_0)$ and $p_1 = \mathcal{N}(x_1|\mu_1, \Sigma_1)$ be Gaussians. Consider any initial discrete Gaussian process $q \in \mathcal{P}_{2,ac}(\mathbb{R}^{D\times(N+2)})$ that has joint distribution $q(x_0, x_1) \in \Pi(p_0, p_1)$:

$$x_{01} \overset{\text{def}}{=} \begin{pmatrix} x_0 \\ x_1 \end{pmatrix}, \quad \mu_{01} \overset{\text{def}}{=} \begin{pmatrix} \mu_0 \\ \mu_1 \end{pmatrix}, \quad \Sigma = \begin{pmatrix} \Sigma_0 & \Sigma_{\text{cov}} \\ \Sigma_{\text{cov}}^T & \Sigma_1 \end{pmatrix}, \quad q(x_0, x_1) \overset{\text{def}}{=} \mathcal{N}(x_{01}|\mu_{01}, \Sigma) \tag{13}$$

where $\Sigma \in \mathbb{R}^{2D\times 2D}$ is positive definite and symmetric and $\Sigma_{\text{cov}}$ is the covariance of $x_0$ and $x_1$. In this case, the result of updates (12) is always a discrete Gaussian processes with specific parameters. To show this, we introduce two auxiliary matrices $U \in \mathbb{R}^{ND\times 2D}$ and $K \in \mathbb{R}^{ND\times ND}$:

$$U \overset{\text{def}}{=} \begin{pmatrix} (1-t_1)I_D & t_1 I_D \\ (1-t_2)I_D & t_2 I_D \\ \vdots & \vdots \\ (1-t_N)I_D & t_N I_D \end{pmatrix}, \quad K \overset{\text{def}}{=} \begin{pmatrix} t_1(1-t_1)I_D & t_1(1-t_2)I_D & \ldots & t_1(1-t_N)I_D \\ t_1(1-t_2)I_D & t_2(1-t_2)I_D & \ldots & t_2(1-t_N)I_D \\ \vdots & \vdots & \ldots & \vdots \\ t_1(1-t_N)I_D & t_2(1-t_N)I_D & \ldots & t_N(1-t_N)I_D \end{pmatrix}$$

Here $I_D$ is an identity matrix with the shape $D \times D$. Below we present updates for both projections.

**Theorem 3.7** (Reciprocal projection of a process whose joint marginal distribution is Gaussian).
*Assume that $q \in \mathcal{P}_{2,ac}(\mathbb{R}^{D \times (N+2)})$ has Gaussian joint distribution $q(x_0, x_1)$ given by (13). Then*

$$[proj_{\mathcal{R}} q](x_{in}, x_0, x_1) = \mathcal{N}(\begin{pmatrix} x_{in} \\ x_{01} \end{pmatrix} | \begin{pmatrix} U\mu_{01} \\ \mu_{01} \end{pmatrix}, \Sigma_R), \quad \Sigma_R \stackrel{def}{=} \begin{pmatrix} \epsilon K + U\Sigma U^T & U\Sigma \\ (U\Sigma)^T & \Sigma \end{pmatrix} \quad (14)$$

**Theorem 3.8** (Markovian projection of a discrete Gaussian process). *Assume that $q \in \mathcal{P}_{2,ac}(\mathbb{R}^{D \times (N+2)})$ is a discrete Gaussian process with $q(x_0, x_1)$ given by (13) and the density*

$$q(x_{in}, x_0, x_1) = \mathcal{N}(\begin{pmatrix} x_{in} \\ x_{01} \end{pmatrix} | \begin{pmatrix} \mu_{in} \\ \mu_{01} \end{pmatrix}, \widetilde{\Sigma}_R), \quad \mu_{in} = (\mu_{t_1}, \dots, \mu_{t_N}),$$

*where $\mu_{in}$ and $\widetilde{\Sigma}_R$ are some parameters of q. Then its Markovian projection is given by:*

$$[proj_{\mathcal{M}} q](x_{in}, x_0, x_1) = q(x_0) \prod_{n=1}^{N+1} q(x_{t_n} | x_{t_{n-1}}), \quad q(x_{t_n} | x_{t_{n-1}}) = \mathcal{N}(x_{t_n} | \widehat{\mu}_{t_n}(x_{t_{n-1}}), \widehat{\Sigma}_{t_n}),$$

$$\widehat{\mu}_{t_n}(x_{t_{n-1}}) = \mu_{t_n} + (\widetilde{\Sigma}_R)_{t_n, t_{n-1}} ((\widetilde{\Sigma}_R)_{t_{n-1}, t_{n-1}})^{-1} (x_{t_{n-1}} - \mu_{t_{n-1}}),$$

$$\widehat{\Sigma}_{t_n} = (\widetilde{\Sigma}_R)_{t_n, t_n} - (\widetilde{\Sigma}_R)_{t_n, t_{n-1}} ((\widetilde{\Sigma}_R)_{t_{n-1}, t_{n-1}})^{-1} ((\widetilde{\Sigma}_R)_{t_n, t_{n-1}})^T.$$

*In turn, the joint distribution $[proj_{\mathcal{M}} q](x_0, x_1)$ is given by*

$$[proj_{\mathcal{M}} q](x_0, x_1) = \mathcal{N}(\begin{pmatrix} x_0 \\ x_1 \end{pmatrix} | \begin{pmatrix} \mu_0 \\ \mu_1 \end{pmatrix}, \begin{pmatrix} \Sigma_0 & \Sigma_{01} \\ (\Sigma_{01})^T & \Sigma_1 \end{pmatrix}), \Sigma_{01}^T = \Big[ \prod_{n=1}^{N+1} (\widetilde{\Sigma}_R)_{t_{n+1}, t_n} ((\widetilde{\Sigma}_R)_{t_n, t_n})^{-1} \Big] \Sigma_0.$$

*Here $(\widetilde{\Sigma}_R)_{t_i, t_j}$ is the submatrix of $\widetilde{\Sigma}_R$ denoting the covariance of $x_{t_i}$ and $x_{t_j}$, while $\Sigma_0$ and $\Sigma_1$ are covariance matrices of $x_0$ and $x_1$, respectively.*

Thus, if we start D-IMF from some discrete process $q^0$ with marginals $q^0(x_0) = p_0(x_0)$, $q^0(x_1) = p_1(x_1)$ and Gaussian $q(x_0, x_1)$, then at each iteration of our D-IMF procedure $q^l$ will be discrete Gaussian process with the same marginals and eventually will converge to $q^*$. In §4.1, we use our derived closed-form to perform an experimental analysis of D-IMF's convergence depending on the number of intermediate time moments $N$ and the value of coefficient $\epsilon$.

### 3.5 Practical Implemetation of D-IMF: ASBM Algorithm

To implement our D-IMF procedure in practice, one should choose the process $q^0$ and implement both discrete Markovian and reciprocal projections. Note that one is usually not interested in the processes' density but only needs the ability to sample endpoints $x_1$ (or trajectories $x_0, x_{t_1}, \dots, x_{t_N}, x_1$) given a starting point $x_0 (= x_{t_0})$. Thus, to solve SB between $p_0(x_0)$ and $p_1(x_1)$ one should choose $q^0$ to have start and end marginals $q^0(x_0) = p_0(x_0)$ and $q^0(x_1) = p(x_1)$ accessible by samples.

**Implementation of the discrete reciprocal projection.** The reciprocal projection (8) of a given discrete process $q(x_0, x_{in}, x_1)$ is easy if one can sample from $q(x_0, x_1)$. To sample from $proj_{\mathcal{R}}(q)$ it is enough to sample first a pair $(x_0, x_1) \sim q(x_0, x_1)$ and then sample intermediate points $x_{t_1}, \dots, x_{t_N}$ from the Brownian bridge $p^{W^\epsilon}(x_{t_1}, \dots, x_{t_N} | x_0, x_1)$. This can be straightforwardly done using the formula (5) where the involved distributions (6) are simple Gaussians which are easy to sample from.

**Implementation of the discrete Markovian projection via DD-GAN.** To find the Markovian projection (10) of a reciprocal process $q \in \mathcal{R}(N)$, one just needs to estimate the transition probabilities between sequential time moments, i.e., the set $\{q(x_{t_n} | x_{t_{n-1}})\}_{n=1}^{N+1}$ and use the starting marginal $[proj_{\mathcal{M}} q](x_0) = q(x_0) = p_0(x_0)$. The natural way to find transition probabilities is to set to parametrize all these distributions as $\{q_\theta(x_{t_n} | x_{t_{n-1}})\}_{n=1}^{N+1}$ and solve

$$\min_\theta \sum_{n=1}^{N+1} \mathbb{E}_{q(x_{t_{n-1}})} D_{adv}(q(x_{t_n} | x_{t_{n-1}}) || q_\theta(x_{t_n} | x_{t_{n-1}})), \quad (15)$$

where $D_{adv}$ is some distance or divergence between probability distributions. In this case, a minimum of such loss is achieved when $q_\theta(x_{t_n} | x_{t_{n-1}}) = q(x_{t_n} | x_{t_{n-1}})$ for each $n \in \{1, 2, \dots, N+1\}$.

We note that a related setting is considered in the Denoising Diffusion GANs (DD-GAN), see [53, Eq. 4]. The difference is in the nature of $q$: there $q$ comes from the standard noising diffusion process, while in our case it is a given reciprocal process. Overall, the authors show that problems like (15)

can be efficiently approached via time-conditioned GANs. Therefore, we naturally pick *DD-GAN approach as the backbone* to learn our discrete Markovian projection and use their best practices.

In short, following DD-GAN, we parameterize $q_\theta(x_{t_n}|x_{t_{n-1}})$ via a time-conditioned generator network $G_\theta(x_{t_{n-1}}, z, t_{n-1})$. As in DD-GAN, we use the non-saturating GAN loss [10] as $D_{\text{adv}}$, which optimizes softened reverse KL-divergence [46]. To optimize this loss, an additional conditional discriminator network $D(x_{t_{n-1}}, x_{t_n}, t_{n-1})$ is needed. We do not recall technical details here as they are the same as in DD-GAN. For further details on DD-GAN learning, we refer to Appendix D.1.

Note that after learning $\{q_\theta(x_{t_n}|x_{t_{n-1}})\}_{n=1}^{N+1}$ the sampling assumes to take sample from $q_0(x_0) = p(x_0)$ and then sample from $\{q_\theta(x_{t_n}|x_{t_{n-1}})\}_{n=1}^{N+1}$. Hence it is guaranteed that $q_0(x_0) = p(x_0)$, but there may be an approximation error in estimating $q_1(x_1) \approx p(x_1)$. This is due to the asymmetry of the definition of Markovian projection, i.e., it can be written in two equivalent ways:

$$\left[\text{proj}_{\mathcal{M}}(q)\right](x_0, x_{t_1}, ..., x_{t_N}, x_1) = q(x_0)\prod_{n=1}^{N+1}q(x_{t_n}|x_{t_{n-1}}) = q(x_1)\prod_{n=1}^{N+1}q(x_{t_{n-1}}|x_{t_n}).$$

Analogously to the implementation of IMF [47, Algorithm 1], we address this asymmetry in our D-IMF by alternatively learning Markovian projection in forward and reverse directions. To learn Markovian projection in the reverse direction, we just need to use starting marginal $[\text{proj}_{\mathcal{M}}q](x_1) = p_1(x_1)$, parametrize $\{q_\eta(x_{t_{n-1}}|x_{t_n})\}_{n=1}^{N+1}$ and solve:

$$\min_\eta \sum_{n=1}^{N+1}\mathbb{E}_{q(x_{t_n})}D_{\text{adv}}\big(q(x_{t_{n-1}}|x_{t_n})||q_\eta(x_{t_{n-1}}|x_{t_n})\big). \tag{16}$$

In this case $q(x_1) = p_1(x_1)$ is guaranteed, while $q(x_0) \approx p_0(x_0)$.

**Implementation of the D-IMF procedure (ASBM algorithm)**. We start with initialization of $q^0$ by the reciprocal process. Depending on the setup we use initialization with the independent coupling, i.e. $q^0(x_0, x_1) = p_0(x_0)p_1(x_1)$ or a minibatch OT coupling [9, 39], see Appendix D.3 for details.

We follow the best practices of IMF [47] and in the Markovian projection steps, we alternately learn models in the direction $p_0 \to p_1$ and in the reverse direction $p_1 \to p_0$ by using functionals (15) and (16) respectively to avoid the accumulation of errors due to the asymmetry in the definition of the Markovian projection. For details, see Appendix D.2. At the reciprocal projection steps, we use the model $q_\theta(x_0, x_{\text{in}}, x_1)$ or $q_\eta(x_0, x_{\text{in}}, x_1)$ learned to approximate $q^{2l+1}$ to sample pair $(x_0, x_1)$ and then sample intermediate points from Brownian bridge. We use the term **outer iteration** $(K)$ for a sequence of two reciprocal projections and two Markovian projections in different directions.

### 3.6 Relation to Prior Works

There exists a variety of algorithms for learning SB based on different underlying principles: dual form entropic optimal transport algorithms [5, 34, 24, 11, 12, 45], iterative proportional fitting (IPF) algorithms [51, 7, 2], bridge matching [49, 29] and iterative Markovian fitting (IMF) algorithms [47, 35, 28, 20], adversarial algorithms [21], etc. We refer to [13] for a benchmark and to [24, Table 1] for a quick survey of many of them. In turn, in our paper, we specifically focus on the advancement of IMF-type algorithms [47, 35] as it they are not only theoretically well-grounded but also closely connected to the rectified flow approach [30] which works well in large-scale generative modeling [32, 54]. Below we discuss the relation of our contributions (§1) to the prior works in IMF [47, 35].

**Theory I**. As we detailed in §2, basic IMF operates with stochastic processes in continuous time and iteratively performs Markovian and reciprocal projections. Our D-IMF procedure (§3) does the same but in the discrete time, so it might *deceptively* seem like our D-IMF is just an approximation of IMF. However, this is a misleading viewpoint. Indeed, the Markovian projection in the discrete time, in general, does not match with the continuous time Markovian projection. Still our D-IMF procedure *provably* converges to SB. Furthermore, D-IMF procedure can theoretically work with just one intermediate time step (when $N = 1$). Overall, its convergence rate varies depending on the number of intermediate points, see §4.1. Naturally, we conjecture that in the limit $N \to \infty$ (when the time steps $t_1, \ldots, t_N$ densely fill $[0, 1]$) our D-IMF behaves the same as IMF since the discrete and continuous Markovian projections start to be close, see discussion in [47, Appendix E].

**Theory II**. In §3.4, we derive the closed-form expression for our D-IMF updates in the Gaussian case. For the continuous IMF, there exists an analogous result [35, §6.1]. However, unlike our result,

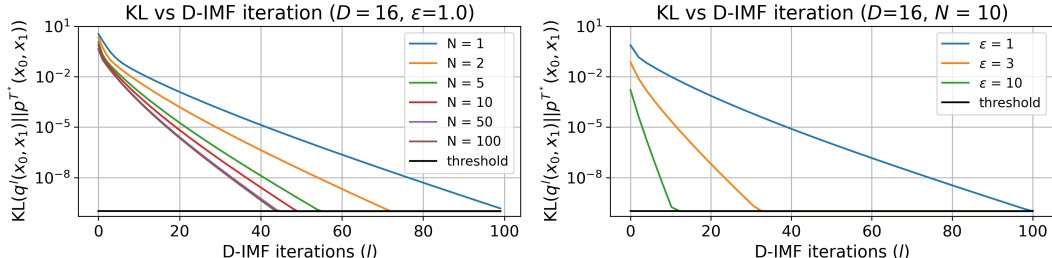

(a) Dependence on the number of time steps $N$.     (b) Dependence on the variance $\epsilon$ of the prior process.

Figure 6: Dependence of convergence of **our** D-IMF procedure on $N$ and $\epsilon$.

that one is not explicit in the sence that it requires solving the matrix-valued ODE [35, Eq. 39] to get the actual projection. The analytical solution is known only when $D = 1$, i.e., 1-dimensional case, see also [47, Appendix D]. In contrast, our Gaussian D-IMF updates work in any dimension $D$.

**Practice.** Default continuous-time IMF [47, 35] in practice is naturally implemented via the Bridge Matching approach which learns an SDE. In our case, at each D-IMF step we learn several transitional probabilities and do this via also well-established adversarial techniques. In this sense, our practical implementation differs – each approach is based on its own backbone – bridge matching vs. adversarial learning – and naturally inherits the benefits/drawbacks of the respective backbone. They are fairly well stated in the discussion of the generative learning trilemma in [53].

## 4 Experiments

We evaluate our adversarial SB matching (**ASBM**) algorithm, which implements our D-IMF procedure on setups with Gaussian distributions (§4.1) for which we have closed form update formulas (§3.4) and real image data distributions (§4.2). We additionally provide results for an illustrative 2D example in Appendix C.1, results for the Colored MNIST dataset in Appendix C.3, and results on the standard SB benchmark in Appendix C.2. The code for our algorithm and all experiments with it is written in Pytorch, is available in the supplementary materials, and will be made public. We provide all the technical details in Appendix D.

### 4.1 Gaussian-to-Gaussian Schrödinger Bridge

We analyze the convergence of our D-IMF procedure depending on the number of intermediate time steps $N \geq 1$ (we use $t_n = n/N + 1$) and the value $\epsilon > 0$ in the Gaussian case. In this case, the static SB solution $p^{T^*}(x_0, x_1)$ is analytically known, see, e.g., [18]. This provides us an opportunity to analyse how fast KL $\left(q^l(x_0, x_1) \| p^{T^*}(x_0, x_1)\right)$ decreases when $l \to \infty$.

We conduct experiments by using our analytical formulas for D-IMF from §3.4. We follow setup from [12] and consider Schrödinger Bridge problem with the dimensionality $D = 16$ and $\epsilon \in \{1, 3, 10\}$ for centered Gaussians $p_0 = \mathcal{N}(0, \Sigma_0)$ and $p_1 = \mathcal{N}(0, \Sigma_1)$. To construct $\Sigma_0$ and $\Sigma_1$, sample their eigenvectors from the uniform distribution on the unit sphere and sample their eigenvalues from the log uniform distribution on $[-\log 2, \log 2]$. We use the same $p_0$ and $p_1$ for all experiments.

We start our D-IMF procedure from the reciprocal process with $q^0(x_0, x_1) = p_0(x_0)p_1(x_1)$, i.e., from the independent joint distribution at times $t = 0, 1$. We present the convergence plots in Figures 6a and 6b. In both plots, we use $10^{-10}$ as a threshold corresponding to the exact matching of distributions to prevent numerical instabilities. We see that our D-IMF procedure empirically shows the exponential rate of convergence in all the cases. As we can see from Figure 6a, the convergence speed dependence on $N$ quickly saturates. Thus, even several time moments, e.g., $N = 5$, provide quick convergence speed. From Figure 6b, we clearly see that the convergence speed is highly influenced by the chosen value of the parameter $\epsilon$. For instance, the transition from $\epsilon = 1$ to $\epsilon = 10$ requires ten times more D-IMF iterations. Thus, this hyperparameter may be important in practice.

### 4.2 Unpaired Image-to-image Translation

To test our approach on real data, we consider the unpaired image-to-image translation setup of learning *male* → *female* faces of Celeba dataset [33]. We use $10\%$ of *male* and *female* images as the test set for evaluation. We train our ASBM algorithm based on the D-IMF procedure with $\epsilon = 1$ and $\epsilon = 10$. Following the best practices of DD-GAN [53], we use $N = 3$, intermediate times $t_1 = \frac{1}{4}, t_2 = \frac{2}{4}, t_3 = \frac{3}{4}$ and $K = 5$ outer iterations of D-IMF. We provide qualitative results and the FID metric [14] on the test set in Figures 7b and 7e. Since we use $N = 3$ intermediate time moments, our algorithm requires only $4$ number of function evaluations (NFE) at the inference stage.

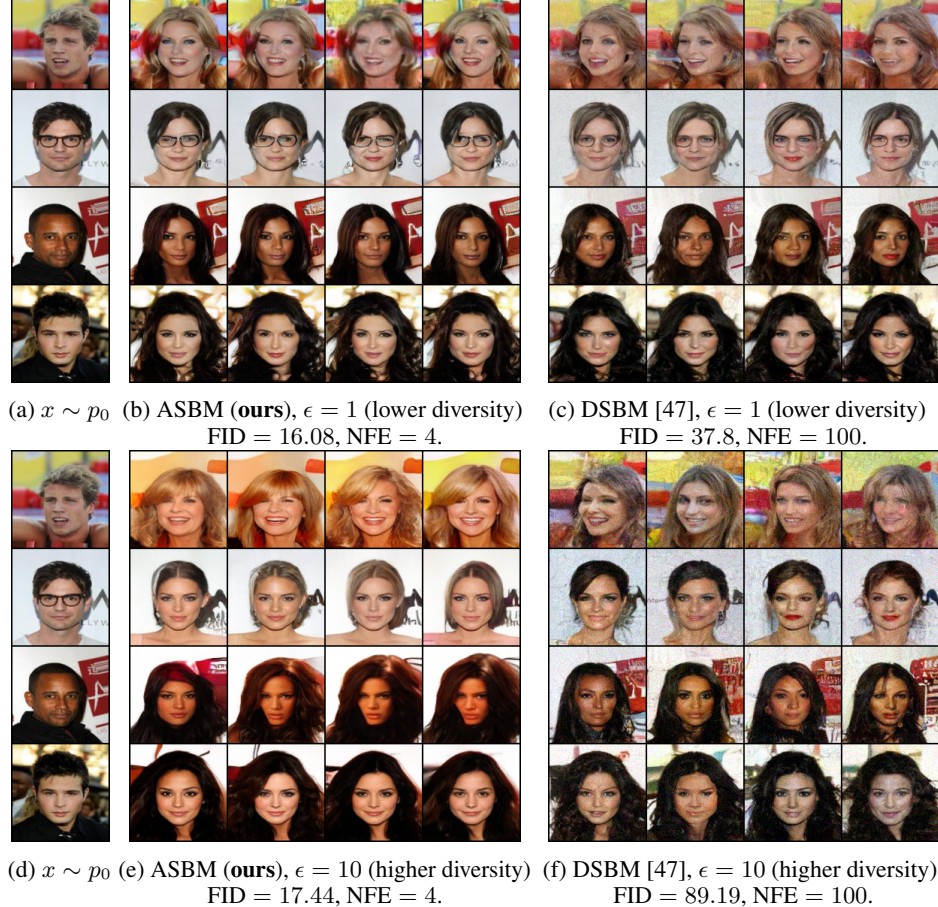

(a) $x \sim p_0$  (b) ASBM (**ours**), $\epsilon = 1$ (lower diversity)  (c) DSBM [47], $\epsilon = 1$ (lower diversity)
FID $= 16.08$, NFE $= 4$.        FID $= 37.8$, NFE $= 100$.

(d) $x \sim p_0$  (e) ASBM (**ours**), $\epsilon = 10$ (higher diversity)  (f) DSBM [47], $\epsilon = 10$ (higher diversity)
FID $= 17.44$, NFE $= 4$.        FID $= 89.19$, NFE $= 100$.

Figure 7: Results of Celeba, *male→female* translation learned with ASBM (ours), and DSBM learned on Celeba dataset with 128 resolution size for $\epsilon \in \{1, 10\}$.

We focus our comparison on the DSBM algorithm based on the IMF-procedure [47] since it is closely related to our method. We train DSBM following the authors [47] and use NFE $= 100$. As well as for ASBM, we use 5 outer iterations of IMF, corresponding to the same number of reciprocal and Markovian projections, but for continuous processes. We use approximately the same number of parameters of neural networks used for models in Markovian projections for ASBM and DSBM (see Appendix D.3). For other details, see Appendix D.4. We present results for DSBM in Figure 7c and Figure 7f. Our algorithm provides better results while using only 4 evaluation steps. Further additional results and measurements for ASBM and DSBM algorithms on the Celeba dataset are presented in Appendix E.

Thus, our D-IMF procedure allows us to solve the Schrödinger Bridge efficiently without learning the time-continuous stochastic process, which in turn speeds up inference by an order of magnitude. This aligns with the results obtained in the Gaussian-to-Gaussian setups about exponentially fast convergence of D-IMF even with several intermediate time moments.

## 5 Discussion

**Potential impact.** Beside the pure speed up of the inference of IMF, we want to point to another great advantage of our developed D-IMF framework. In the continuous IMF, one is forced to do Markovian projection via time-consuming learning of continuous-time SDEs (using procedures like bridge matching). In our D-IMF framework, one needs to **learn several transition probabilities**. We do this via adversarial learning [10], but actually this can be done **using** almost **any other generative modeling technique** (moment matching [26], normalizing flows [23, 41], energy-based models [56], score-based models [48], etc.). We believe that this observation opens great possibilities for ML community to further explore and improve generative modeling algorithms based on Schrödinger Bridges, Markovian projections (bridge matching) and related techniques, e.g., flow matching [27].

Limitations and broader impact are discussed in Appendix A.

## Acknowledgements

The work was supported by the Analytical center under the RF Government (subsidy agreement 000000D730321P5Q0002, Grant No. 70-2021-00145 02.11.2021).

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

# A  Limitations and Future Work

**Adversarial training**. It is a generic knowledge that the adversarial training may be non trivial to conduct due to instabilities, mode collapse and related issues. Fortunately, our ASBM algorithm relies on the already well-established and carefully tuned DD-GAN [53] technique as a backbone. The latter is specifically designed to address many such limitations and is known to score good metrics in generative modeling.

**Theoretical convergence rate**. We derive the generic convergence result for our D-IMF procedure (Theorem 3.6) but without the particular convergence rate. Empirically we observe the exponentially fast convergence (§4.1), but theoretically proving this rate is an important task for the future work.

**Broader Impact.**  This paper presents work whose goal is to advance the field of Artificial Intelligence, Machine Learning and Generative Modeling. There are many potential societal consequences of our work, none which we feel must be specifically highlighted here.

# B  Proofs

Here we provide the proof of our theoretical results one-by-one. Additionally, we introduce and prove several auxiliary results to simplify the derivation of the main results.

## B.1  Proofs for Statements in Section 3.2

In our view, the proof of our main Theorem here is the most interesting and insightful (among all the proofs in the paper) as it uses some tricky mathematics, especially in its stage 2. In turn, stage 1 of the proof is inspired by the recent insights of [13, Theorem 3.2] about the characterization of static Schrödinger Bridge solutions [25] and manipulations with KL for SB in [24, 11].

*Proof of Theorem 3.1.*  We split the proof in 2 stages. The 1st is auxiliary for the 2nd.

**Stage 1.** Here we show that if some $q(x_0, x_1) \in \mathcal{P}_{2,ac}(\mathbb{R}^{D \times 2})$ with marginals $p_0(x_0) = q(x_0)$ and $p_1(x_1) = q(x_1)$ has the density in the form

$$q(x_0, x_1) = q(x_0)\widehat{C}(x_0) \exp\left(-\frac{||x_1 - x_0||^2}{2\epsilon}\right)\widehat{\phi}(x_1),$$

then it solves the Static SB between distributions $p_0(x_0)$ and $p_1(x_1)$. It is known [25], that the solution $q^*(x_0, x_1) \stackrel{\text{def}}{=} p^{T^*}(x_0, x_1)$ of Static SB between $p_0$ and $p_1$ has the density:

$$q^*(x_0, x_1) = \psi^*(x_0) \exp\left(-\frac{||x_1 - x_0||^2}{2\epsilon}\right)\phi^*(x_1).$$

Hence, the conditional density $q^*(x_1|x_0)$ is expressed as:

$$q^*(x_1|x_0) = \underbrace{\frac{\psi^*(x_0)}{p_0(x_0)}}_{\stackrel{\text{def}}{=}C^*(x_0)} \exp\left(-\frac{||x_1 - x_0||^2}{2\epsilon}\right)\phi^*(x_1) = C^*(x_0)\exp\left(-\frac{||x_1 - x_0||^2}{2\epsilon}\right)\phi^*(x_1).$$

Thus, both $q(x_0, x_1)$ and $q^*(x_0, x_1)$ have their densities in the same functional form and the same marginals $q(x_0) = q^*(x_0) = p_0(x_0)$ and $q(x_1) = q^*(x_1) = p_1(x_1)$. However, we want to prove that in this case $q(x_0, x_1)$ and $q^*(x_0, x_1)$ are equal, i.e., KL $(q^*\|q) = 0$.

$$\text{KL}\left(q^*(x_0, x_1)\|q(x_0, x_1)\right) = \int \log \frac{q^*(x_0, x_1)}{q(x_0, x_1)} q^*(x_0, x_1)dx_0 dx_1 =$$

$$\int \log \frac{p_0(x_0)q^*(x_1|x_0)}{p_0(x_0)q(x_1|x_0)} q^*(x_0, x_1)dx_0 dx_1 =$$

$$\int \log \frac{C^*(x_0)\exp\left(-\frac{||x_1 - x_0||^2}{2\epsilon}\right)\phi^*(x_1)}{\widehat{C}(x_0)\exp\left(-\frac{||x_1 - x_0||^2}{2\epsilon}\right)\widehat{\phi}(x_1)} q^*(x_0, x_1)dx_0 dx_1 =$$

$$\int (\log \frac{C^*(x_0)}{\widehat{C}(x_0)} + \log \frac{\phi^*(x_1)}{\widehat{\phi}(x_1)}) q^*(x_0, x_1) dx_0 dx_1 =$$

$$\int \log \frac{C^*(x_0)}{\widehat{C}(x_0)} q^*(x_0, x_1) dx_0 dx_1 + \int \log \frac{\phi^*(x_1)}{\widehat{\phi}(x_1)} q^*(x_0, x_1) dx_0 dx_1 =$$

$$\int \log \frac{C^*(x_0)}{\widehat{C}(x_0)} p_0(x_0) dx_0 + \int \log \frac{\phi^*(x_1)}{\widehat{\phi}(x_1)} p_1(x_1) dx_1 =$$

$$\int \log \frac{C^*(x_0)}{\widehat{C}(x_0)} q(x_0, x_1) dx_0 dx_1 + \int \log \frac{\phi^*(x_1)}{\widehat{\phi}(x_1)} q(x_0, x_1) dx_0 dx_1 =$$

$$\int (\log \frac{C^*(x_0)}{\widehat{C}(x_0)} + \log \frac{\phi^*(x_1)}{\widehat{\phi}(x_1)}) q(x_0, x_1) dx_0 dx_1 =$$

$$\int \log \frac{C^*(x_0)\phi^*(x_1)}{\widehat{C}(x_0)\widehat{\phi}(x_1)} q(x_0, x_1) dx_0 dx_1 =$$

$$\int \log \frac{C^*(x_0) \exp\left(-\frac{||x_1-x_0||^2}{2\epsilon}\right)\phi^*(x_1)}{\widehat{C}(x_0) \exp\left(-\frac{||x_1-x_0||^2}{2\epsilon}\right)\widehat{\phi}(x_1)} q(x_0, x_1) dx_0 dx_1 =$$

$$\int \log \frac{q^*(x_1|x_0)}{q(x_1|x_0)} q(x_0, x_1) dx_0 dx_1 =$$

$$\int \log \frac{p_0(x_0)q^*(x_1|x_0)}{p_0(x_0)q(x_1|x_0)} q(x_0, x_1) dx_0 dx_1 =$$

$$\int \log \frac{q^*(x_0, x_1)}{q(x_0, x_1)} q(x_0, x_1) dx_0 dx_1 =$$

$$-\int \log \frac{q(x_0, x_1)}{q^*(x_0, x_1)} q(x_0, x_1) dx_0 dx_1 = -\text{KL}\left(q(x_0, x_1) \| q^*(x_0, x_1)\right).$$

Thus, $\text{KL}\left(q^*\|q\right) = -\text{KL}\left(q\|q^*\right)$. Since the KL divergence is non-negative, we derive that $q = q^*$.

**Stage 2.** In this stage, we prove the theorem itself. First, if $N > 1$, i.e., there is more than one intermediate time moment, we integrate $q(x_0, x_{t_1}, \ldots, x_{t_N}, x_1)$ over all intermediate time moments except $t_1$. On the one hand, we get

$$q(x_0, x_{t_1}, x_1) = \int q(x_0, x_{t_1}, \ldots, x_{t_N}, x_1) dx_{t_2} \ldots dx_{t_N} =$$

$$\int p^{W^\epsilon}(x_{t_1}, \ldots, x_{t_N}|x_0, x_1) q(x_0, x_1) dx_{t_2} \ldots dx_{t_N} = p^{W^\epsilon}(x_{t_1}|x_0, x_1) q(x_1, x_0). \qquad (17)$$

On the other hand, we derive

$$q(x_0, x_{t_1}, x_1) = \int q(x_0) q(x_{t_1}|x_0) \overbrace{\prod_{n=2}^{N+1} q(x_{t_n}|x_{t_{n-1}})}^{=q(x_{t_2}, \ldots, x_{t_N}, x_1|x_{t_1})} dx_{t_2} \ldots dx_{t_N} = q(x_0) q(x_{t_1}|x_0) q(x_1|x_{t_1}). \quad (18)$$

Combining (17) and (18) yields

$$q(x_0) q(x_{t_1}|x_0) q(x_1|x_{t_1}) = q(x_0, x_{t_1}, x_1) = p^{W^\epsilon}(x_{t_1}|x_0, x_1) q(x_1, x_0). \qquad (19)$$

Note that if $N = 1$, then we already have (19) from the conditions of the theorem. Therefore,

$$p^{W^\epsilon}(x_{t_1}|x_0, x_1) q(x_1, x_0) = q(x_0) q(x_{t_1}|x_0) q(x_1|x_{t_1})$$

$$p^{W^\epsilon}(x_{t_1}|x_0, x_1) q(x_1|x_0) \cancel{q(x_0)} = \cancel{q(x_0)} q(x_{t_1}|x_0) q(x_1|x_{t_1}).$$

From now on, we are interested only in 3 time moments: $t_0 = 0, t_1$ and $t_{N+1} = 1$. To simplify the notation, we will write $t$ instead of $t_2$ in the following proof. We take the logarithm and get

$$\log q(x_1|x_0) = \log q(x_t|x_0) + \log q(x_1|x_t) - \log p^{W^\epsilon}(x_t|x_0, x_1)$$

Then, we use the formula for the Brownian Bridge density:

$$\log q(x_1|x_0) = \log q(x_t|x_0) + \log q(x_1|x_t) - C + \frac{1}{2\epsilon t(1-t)}||x_t - (tx_1 + (1-t)x_0)||^2 =$$

$$\log q(x_t|x_0) + \log q(x_1|x_t) - C +$$

$$\frac{1}{2\epsilon t(1-t)}(||x_t||^2 + ||tx_1||^2 + ||(1-t)x_0||^2 - 2tx_t^T x_1 - 2(1-t)x_t^T x_0 + 2t(1-t)x_0^T x_1) =$$

$$\log q(x_t|x_0) + \log q(x_1|x_t) - C +$$

$$\frac{||x_t||^2}{2\epsilon t(1-t)} + \frac{||(1-t)x_0||^2}{2\epsilon t(1-t)} + \frac{||tx_1||^2}{2\epsilon t(1-t)} - \frac{x_t^T x_1}{\epsilon(1-t)} - \frac{x_t^T x_0}{\epsilon t} + \frac{x_0^T x_1}{\epsilon} =$$

$$\underbrace{\log q(x_1|x_t) + \frac{||x_t||^2}{2\epsilon t(1-t)} + \frac{||tx_1||^2}{2\epsilon t(1-t)} - \frac{x_t^T x_1}{\epsilon(1-t)} - C +}_{\overset{\text{def}}{=} f_1(x_t, x_1)}$$

$$\underbrace{\frac{||(1-t)x_0||^2}{2\epsilon t(1-t)} + \log q(x_t|x_0) - \frac{x_t^T x_0}{\epsilon t}}_{\overset{\text{def}}{=} f_2(x_t, x_0)} + \frac{x_0^T x_1}{\epsilon}.$$

Thus,

$$\log q(x_1|x_0) = f_1(x_t, x_1) + f_2(x_t, x_0) + \frac{x_0^T x_1}{\epsilon},$$

$$\underbrace{\log q(x_1|x_0) - \frac{x_0^T x_1}{\epsilon}}_{\overset{\text{def}}{=} f_3(x_0, x_1)} = f_1(x_t, x_1) + f_2(x_t, x_0),$$

$$f_3(x_0, x_1) = f_1(x_t, x_1) + f_2(x_t, x_0). \tag{20}$$

Below, we prove that $f_3(x_0, x_1) = g_1(x_0) + g_2(x_1)$ for some functions $g_1$ and $g_2$. We note that

$$f_3(x_0, 0) = f_1(x_t, 0) + f_2(x_t, x_0) \qquad \Rightarrow \qquad f_2(x_t, x_0) = f_3(x_0, 0) - f_1(x_t, 0). \tag{21}$$

We substitute (21) to (20):

$$f_3(x_0, x_1) = f_1(x_t, x_1) + \underbrace{f_3(x_0, 0) - f_1(x_t, 0)}_{= f_2(x_t, x_0)}$$

$$f_3(x_0, x_1) - f_3(x_0, 0) = f_1(x_t, x_1) - f_1(x_t, 0). \tag{22}$$

Since there is no dependence on $x_0$ in the right part of (22), we conclude that $f_3(x_0, x_1) - f_3(x_0, 0)$ is a function of only $x_1$. We define $g_1(x_1) \overset{\text{def}}{=} f_3(x_0, x_1) - f_3(x_0, 0)$ and $g_2(x_0) \overset{\text{def}}{=} f_3(x_0, 0)$. Now we have the desired result:

$$f_3(x_0, x_1) = g_1(x_1) + f_3(x_0, 0) = g_1(x_1) + g_2(x_0). \tag{23}$$

Thus,

$$f_3(x_0, x_1) = \log q(x_1|x_0) - \frac{x_0^T x_1}{\epsilon} = g_1(x_1) + g_2(x_0).$$

Now, we can use this result about the separation of variables together with the result from the first **stage** to conclude the proof of the theorem.

$$\log q(x_1|x_0) = g_1(x_1) + g_2(x_0) + \frac{x_0^T x_1}{\epsilon} =$$

$$g_1(x_1) + \frac{||x_1||^2}{2\epsilon} + g_2(x_0) + \frac{||x_0||^2}{2\epsilon} - \frac{||x_0 - x_1||^2}{2\epsilon},$$

$$q(x_1|x_0) = \underbrace{\exp\left(g_2(x_0) + \frac{||x_0||^2}{2\epsilon}\right)}_{\overset{\text{def}}{=} C(x_0)} \exp\left(-\frac{||x_0 - x_1||^2}{2\epsilon}\right) \underbrace{\exp\left(g_1(x_1) + \frac{||x_1||^2}{2\epsilon}\right)}_{\overset{\text{def}}{=} \phi(x_1)} =$$

$$q(x_1|x_0) = C(x_0)\exp\left(-\frac{||x_0 - x_1||^2}{2\epsilon}\right)\phi(x_1),$$

$$q(x_0, x_1) = q(x_0)C(x_0)\exp\left(-\frac{||x_0 - x_1||^2}{2\epsilon}\right)\phi(x_1).$$

Hence, from the first **stage** of this proof it follows that $q(x_0, x_1)$ is the solution to the Static SB between $q(x_0) = p_0(x_0)$ and $q(x_1) = p_1(x_1)$ with the coefficient $\epsilon$. That is, $p^{T^*}(x_0, x_1) = q(x_0, x_1)$. Since $q(x_{t_1}, \ldots, x_{t_N}|x_0, x_1) = p^{W^\epsilon}(x_{t_1}, \ldots, x_{t_N}|x_0, x_1)$ by the assumptions of the current theorem, we also conclude that $q(x_0, x_{t_1}, \ldots, x_{t_N}, x_1) = p^{T^*}(x_0, x_{t_1}, \ldots, x_{t_N}, x_1)$, i.e., the discrete processes coincide. $\qquad\square$

## B.2 Proofs for Statements in Section 3.3

The logic of our justification of D-IMF for discrete processes generally follows the respective logic of the justification of IMF for continuous stochastic processes [47].

*Proof of Proposition 3.3.* The mild assumption here consists in the existence of at least one process $r \in \mathcal{R}(N)$ for which KL $(q\|r) < \infty$. The reciprocal process $r \in \mathcal{R}(N)$ has its density in the form $r(x_0, x_{t_1}, \ldots, x_{t_N}, x_1) = p^{W^\epsilon}(x_{t_1}, \ldots, x_{t_N}|x_0, x_1)r(x_0, x_1)$ (see (7)). Thus, we need to optimize only the part $r(x_0, x_1)$. Below we show, that $r(x_0, x_1)$ should be equal $q(x_0, x_1)$ to minimize the functional.

$$\text{KL}(q\|r) = \int \log\frac{q(x_0, x_{\text{in}}, x_1)}{r(x_0, x_{\text{in}}, x_1)}q(x_0, x_{\text{in}}, x_1)dx_0 dx_{\text{in}} dx_1 =$$

$$\int \log\frac{q(x_{\text{in}}|x_0, x_1)q(x_0, x_1)}{\underbrace{r(x_{\text{in}}|x_0, x_1)}_{p^{W^\epsilon}(x_{\text{in}}|x_0, x_1)}r(x_0, x_1)}q(x_0, x_{\text{in}}, x_1)dx_0 dx_{\text{in}} dx_1 =$$

$$\underbrace{\int \log\frac{q(x_{\text{in}}|x_0, x_1)}{p^{W^\epsilon}(x_{\text{in}}|x_0, x_1)}q(x_0, x_{\text{in}}, x_1)dx_0 dx_{\text{in}} dx_1}_{=\text{Const}} + \int \log\frac{q(x_0, x_1)}{r(x_0, x_1)}q(x_0, x_{\text{in}}, x_1)dx_0 dx_{\text{in}} dx_1 =$$

$$\text{Const} + \underbrace{\int \log\frac{q(x_0, x_1)}{r(x_0, x_1)}q(x_0, x_1)dx_0 dx_1}_{=\text{KL}(q(x_0,x_1)\|r(x_0,x_1))} = \text{Const} + \text{KL}(q(x_0, x_1)\|r(x_0, x_1))$$

Hence, $\text{proj}_{\mathcal{R}}(q) = \arg\min_{r\in\mathcal{R}(N)}\text{KL}(q\|r) = p^{W^\epsilon}(x_{\text{in}}|x_0, x_1)q(x_0, x_1)$. $\qquad\square$

*Proof of Proposition 3.5.* Similar to the previous proposition, the mild assumption here consists in the existence of at least one process $m \in \mathcal{M}(N)$ for which KL $(q\|m) < \infty$. This proof is a bit more technical than for the reciprocal projection. We need to define new notation $x_{t_n, t_{n-1}}^{\text{not}} = (x_{t_0}, \ldots, x_{t_{n-2}}, x_{t_{n+1}}, \ldots, x_{t_{N+1}})$ for a vector of variables for all time moment except two time moments $t_n$ and $t_{n-1}$.

$$\text{KL}(q\|m) = \int \log\frac{q(x_0, x_{\text{in}}, x_1)}{m(x_0, x_{\text{in}}, x_1)}q(x_0, x_{\text{in}}, x_1)dx_0 dx_{\text{in}} dx_1 =$$

$$\int \log\frac{q(x_0, x_{\text{in}}, x_1)}{m(x_0)\prod_{n=1}^{N+1}m(x_{t_n}|x_{t_{n-1}})}q(x_0, x_{\text{in}}, x_1)dx_0 dx_{\text{in}} dx_1 =$$

$$\int \log\frac{q(x_0)}{m(x_0)}q(x_0, x_{\text{in}}, x_1)dx_0 dx_{\text{in}} dx_1 + \int \log\frac{q(x_{\text{in}}, x_1|x_0)}{\prod_{n=1}^{N+1}m(x_{t_n}|x_{t_{n-1}})}q(x_0, x_{\text{in}}, x_1)dx_0 dx_{\text{in}} dx_1 =$$

$$\underbrace{\int \log\frac{q(x_0)}{m(x_0)}q(x_0)dx_0}_{\text{KL}(q(x_0)\|m(x_0))} + \int \log\frac{q(x_{\text{in}}, x_1|x_0)}{\prod_{n=1}^{N+1}m(x_{t_n}|x_{t_{n-1}})}q(x_0, x_{\text{in}}, x_1)dx_0 =$$

$$\mathrm{KL}\left(q(x_0)\|m(x_0)\right) + \int \log \frac{q(x_{\mathrm{in}}, x_1|x_0)}{\prod_{n=1}^{N+1} m(x_{t_n}|x_{t_{n-1}})} q(x_0, x_{\mathrm{in}}, x_1) dx_0 dx_{\mathrm{in}} dx_1 + \quad (24)$$

$$\underbrace{N \int \log \frac{q(x_0, x_{\mathrm{in}}, x_1)}{q(x_0, x_{\mathrm{in}}, x_1)} q(x_0, x_{\mathrm{in}}, x_1) dx_0 dx_{\mathrm{in}} dx_1}_{=0} + \underbrace{\int \log \frac{q(x_0)}{q(x_0)} q(x_0, x_{\mathrm{in}}, x_1) dx_0 dx_{\mathrm{in}} x_1}_{=0} = \quad (25)$$

$$\mathrm{KL}\left(q(x_0)\|m(x_0)\right) + \int \log \frac{\prod_{n=1}^{N+1} q(x_0, x_{\mathrm{in}}, x_1)}{\prod_{n=1}^{N+1} m(x_{t_n}|x_{t_{n-1}})} q(x_0, x_{\mathrm{in}}, x_1) dx_0 dx_{\mathrm{in}} dx_1 -$$

$$\left(N \int \log q(x_0, x_{\mathrm{in}}, x_1) q(x_0, x_{\mathrm{in}}, x_1) dx_0 dx_{\mathrm{in}} dx_1 + \int \log q(x_0) q(x_0) dx_0 dx_{\mathrm{in}} dx_1\right) =$$

$$\mathrm{KL}\left(q(x_0)\|m(x_0)\right) + \sum_{n=1}^{N+1} \int \log \frac{q(x_0, x_{\mathrm{in}}, x_1)}{m(x_{t_n}|x_{t_{n-1}})} q(x_0, x_{\mathrm{in}}, x_1) dx_0 dx_{\mathrm{in}} dx_1 -$$

$$\underbrace{\left(N \int \log q(x_0, x_{\mathrm{in}}, x_1) q(x_0, x_{\mathrm{in}}, x_1) dx_0 dx_{\mathrm{in}} dx_1 + \int \log q(x_0) q(x_0) dx_0 dx_{\mathrm{in}} dx_1\right)}_{\stackrel{\text{def}}{=} C_1} =$$

$$\mathrm{KL}\left(q(x_0)\|m(x_0)\right) - C_1 +$$

$$\sum_{n=1}^{N+1} \int \log \frac{q(x_{t_n}|x_{t_{n-1}}) q(x_{t_{n-1}}) q(x_{t_n, t_{n-1}}^{\mathrm{not}}|x_{t_n}, x_{t_{n-1}})}{m(x_{t_n}|x_{t_{n-1}})} q(x_0, x_{\mathrm{in}}, x_1) dx_0 dx_{\mathrm{in}} dx_1 =$$

$$\mathrm{KL}\left(q(x_0)\|m(x_0)\right) - C_1 + \underbrace{\sum_{n=1}^{N+1} \int \log \left(q(x_{t_{n-1}}) q(x_{t_n, t_{n-1}}^{\mathrm{not}}|x_{t_n}, x_{t_{n-1}})\right) q(x_0, x_{\mathrm{in}}, x_1) dx_0 dx_{\mathrm{in}} dx_1}_{\stackrel{\text{def}}{=} C_2} +$$

$$\sum_{n=1}^{N+1} \int \log \frac{q(x_{t_n}|x_{t_{n-1}})}{m(x_{t_n}|x_{t_{n-1}})} q(x_0, x_{\mathrm{in}}, x_1) dx_0 dx_{\mathrm{in}} dx_1 =$$

$$\mathrm{KL}\left(q(x_0)\|m(x_0)\right) + C_2 + \sum_{n=1}^{N+1} \Big( \underbrace{\int \log \frac{q(x_{t_n}|x_{t_{n-1}})}{m(x_{t_n}|x_{t_{n-1}})} q(x_{t_n}|x_{t_{n-1}}) dx_{t_n}}_{\mathrm{KL}\left(q(x_{t_n}|x_{t_{n-1}})\|m(x_{t_n}|x_{t_{n-1}})\right)} \Big) q(x_{t_{n-1}}) dx_{t_{n-1}} =$$

$$\mathrm{KL}\left(q(x_0)\|m(x_0)\right) + \sum_{n=1}^{N+1} \int \mathrm{KL}\left(q(x_{t_n}|x_{t_{n-1}})\|m(x_{t_n}|x_{t_{n-1}})\right) q(x_{t_{n-1}}) dx_{t_{n-1}} + C_2.$$

In the line (25), we add terms equal to zero, to match each $m(x_{t_n}|x_{t_{n-1}})$ by the separate term $q(x_0, x_{\mathrm{in}}, x_1)$ in the line (25). We need it to as we want to place each term $m(x_{t_n}|x_{t_{n-1}})$ in the separate KL-divergence in the final expression. Hence, the minimizer of the objective $m^* \in \mathcal{M}(N)$ has $m^*(x_0) = q(x_0)$ and all transitional distributions $m^*(x_{t_n}|x_{t_{n-1}}) = q(x_{t_n}|x_{t_{n-1}})$, i.e. is given by

$$m^*(x_0, x_{\mathrm{in}}, x_1) = [\mathrm{proj}_{\mathcal{M}}(q)](x_0, x_{\mathrm{in}}, x_1) = q(x_0) \prod_{n=1}^{N+1} q(x_{t_n}|x_{t_{n-1}}).$$

$\square$

**Proposition B.1** (Pythagorean theorems for projections). *Assume that $r \in \mathcal{R}(N)$ and $m \in \mathcal{M}(N)$. If $KL\left(r\|m\right) < \infty$ and $KL\left(r\|proj_{\mathcal{M}}(r)\right) < \infty$, then*

$$KL\left(r\|m\right) = KL\left(r\|proj_{\mathcal{M}}(r)\right) + KL\left(proj_{\mathcal{M}}(r)\|m\right) \qquad (26)$$

*and if $KL\left(m\|r\right) < \infty$, $KL\left(m\|proj_{\mathcal{R}}(m)\right) < \infty$ then*

$$KL\left(m\|r\right) = KL\left(m\|proj_{\mathcal{R}}(m)\right) + KL\left(proj_{\mathcal{R}}(m)\|r\right)$$

*Proof of Proposition B.1.* Before proving the first equation (26) we prove the additional property of $r \in \mathcal{R}(N)$ for any $n \in [1, 2, \dots, N+1]$:

$$[\mathrm{proj}_{\mathcal{M}} r](x_{t_n}, x_{t_{n-1}}) = r(x_{t_n}, x_{t_{n-1}}).$$

$$[\text{proj}_{\mathcal{M}} r](x_{t_n}, x_{t_{n-1}}) = [\text{proj}_{\mathcal{M}} r](x_{t_n}|x_{t_{n-1}})[\text{proj}_{\mathcal{M}} r](x_{t_n}) = r(x_{t_n}|x_{t_{n-1}})r(x_{t_n}). \qquad (27)$$

Since $[\text{proj}_{\mathcal{M}} r](x_{t_n}|x_{t_{n-1}}) = r(x_{t_n}|x_{t_{n-1}})$ by the definition and since Markovian projection preserve all intermediate time marginals. Now we prove the first equation (26).

$$\text{KL}(r\|m) = \int \log \frac{r(x_0, x_{\text{in}}, x_1)}{m(x_0, x_{\text{in}}, x_1)} r(x_0, x_{\text{in}}, x_1) dx_0 dx_{\text{in}} dx_1 =$$

$$\int \log \frac{r(x_0, x_{\text{in}}, x_1)}{m(x_0, x_{\text{in}}, x_1)} r(x_0, x_{\text{in}}, x_1) dx_0 dx_{\text{in}} dx_1 +$$

$$\underbrace{\int \log \frac{[\text{proj}_{\mathcal{M}}(r)](x_0, x_{\text{in}}, x_1)}{[\text{proj}_{\mathcal{M}}(r)](x_0, x_{\text{in}}, x_1)} r(x_0, x_{\text{in}}, x_1) dx_0 dx_{\text{in}} dx_1}_{=0} =$$

$$\underbrace{\int \log \frac{r(x_0, x_{\text{in}}, x_1)}{[\text{proj}_{\mathcal{M}}(r)](x_0, x_{\text{in}}, x_1)} r(x_0, x_{\text{in}}, x_1) dx_0 dx_{\text{in}} dx_1}_{\text{KL}(r\|\text{proj}_{\mathcal{M}}(r))} +$$

$$\int \log \frac{[\text{proj}_{\mathcal{M}}(r)](x_0, x_{\text{in}}, x_1)}{m(x_0, x_{\text{in}}, x_1)} r(x_0, x_{\text{in}}, x_1) dx_0 dx_{\text{in}} dx_1 =$$

$$\text{KL}(r\|\text{proj}_{\mathcal{M}}(r)) + \int \log \frac{[\text{proj}_{\mathcal{M}}(r)](x_0) \prod_{n=1}^{N+1} [\text{proj}_{\mathcal{M}}(r)](x_{t_n}|x_{t_{n-1}})}{m(x_0) \prod_{n=1}^{N+1} m(x_{t_n}|x_{t_{n-1}})} r(x_0, x_{\text{in}}, x_1) dx_0 dx_{\text{in}} dx_1 =$$

$$\text{KL}(r\|\text{proj}_{\mathcal{M}}(r)) + \text{KL}([\text{proj}_{\mathcal{M}}(r)](x_0)\|m(x_0)) +$$

$$\sum_{n=1}^{N+1} \int \log \frac{[\text{proj}_{\mathcal{M}}(r)](x_{t_n}|x_{t_{n-1}})}{m(x_{t_n}|x_{t_{n-1}})} r(x_0, x_{\text{in}}, x_1) dx_0 dx_{\text{in}} dx_1 =$$

$$\text{KL}(r\|\text{proj}_{\mathcal{M}}(r)) + \text{KL}([\text{proj}_{\mathcal{M}}(r)](x_0)\|m(x_0)) +$$

$$\sum_{n=1}^{N+1} \int \log \frac{[\text{proj}_{\mathcal{M}}(r)](x_{t_n}|x_{t_{n-1}})}{m(x_{t_n}|x_{t_{n-1}})} \underbrace{r(x_{t_n}, x_{t_{n-1}})}_{=[\text{proj}_{\mathcal{M}}(r)](x_{t_n}, x_{t_{n-1}})} dx_{t_n} dx_{t_{n-1}} =$$

$$\text{KL}(r\|\text{proj}_{\mathcal{M}}(r)) + \text{KL}([\text{proj}_{\mathcal{M}}(r)](x_0)\|m(x_0)) +$$

$$\sum_{n=1}^{N+1} \int \log \frac{[\text{proj}_{\mathcal{M}}(r)](x_{t_n}|x_{t_{n-1}})}{m(x_{t_n}|x_{t_{n-1}})} [\text{proj}_{\mathcal{M}}(r)](x_{t_n}, x_{t_{n-1}}) dx_{t_n} dx_{t_{n-1}} =$$

$$\text{KL}(r\|\text{proj}_{\mathcal{M}}(r)) + \text{KL}([\text{proj}_{\mathcal{M}}(r)](x_0)\|m(x_0)) +$$

$$\sum_{n=1}^{N+1} \int \log \frac{[\text{proj}_{\mathcal{M}}(r)](x_{t_n}|x_{t_{n-1}})}{m(x_{t_n}|x_{t_{n-1}})} [\text{proj}_{\mathcal{M}}(r)](x_0, x_{\text{in}}, x_1) dx_0 dx_{\text{in}} dx_1 =$$

$$\text{KL}(r\|\text{proj}_{\mathcal{M}}(r)) + \underbrace{\int \log \frac{[\text{proj}_{\mathcal{R}}](q)(x_0)}{m(x_0)} [\text{proj}_{\mathcal{R}}](q)(x_0) dx_0}_{=\text{KL}([\text{proj}_{\mathcal{M}}(r)](x_0)\|m(x_0))} +$$

$$\int \log \frac{\prod_{n=1}^{N+1} [\text{proj}_{\mathcal{M}}(r)](x_{t_n}|x_{t_{n-1}})}{\prod_{n=1}^{N+1} m(x_{t_n}|x_{t_{n-1}})} [\text{proj}_{\mathcal{M}}(r)](x_0, x_{\text{in}}, x_1) dx_0 dx_{\text{in}} dx_1 =$$

$$\text{KL}(r\|\text{proj}_{\mathcal{M}}(r)) + \underbrace{\int \log \frac{[\text{proj}_{\mathcal{M}}(r)](x_0, x_{\text{in}}, x_1)}{m(x_0, x_{\text{in}}, x_1)} [\text{proj}_{\mathcal{M}}(r)](x_0, x_{\text{in}}, x_1) dx_0 dx_{\text{in}} dx_1}_{\text{KL}(\text{proj}_{\mathcal{M}}(r)\|m)} =$$

$$\text{KL}(r\|\text{proj}_{\mathcal{M}}(r)) + \text{KL}(\text{proj}_{\mathcal{M}}(r)\|m).$$

That concludes the proof of the first equation (26). The proof for the second equation (27) is similar.

$$\text{KL}(m\|r) = \int \log \frac{m(x_0, x_{\text{in}}, x_1)}{r(x_0, x_{\text{in}}, x_1)} m(x_0, x_{\text{in}}, x_1) dx_0 dx_{\text{in}} dx_1 +$$

$$\int \log \underbrace{\frac{[\text{proj}_{\mathcal{R}}(m)](x_0, x_{\text{in}}, x_1)}{[\text{proj}_{\mathcal{R}}(m)](x_0, x_{\text{in}}, x_1)}}_{=0} m(x_0, x_{\text{in}}, x_1) dx_0 dx_{\text{in}} dx_1 =$$

$$\underbrace{\int \log \frac{m(x_0, x_{\text{in}}, x_1)}{[\text{proj}_{\mathcal{R}}(m)](x_0, x_{\text{in}}, x_1)} m(x_0, x_{\text{in}}, x_1) dx_0 dx_{\text{in}} dx_1}_{\text{KL}(m\|\text{proj}_{\mathcal{R}}(m))} +$$

$$\int \log \frac{[\text{proj}_{\mathcal{R}}(m)](x_0, x_{\text{in}}, x_1)}{r(x_0, x_{\text{in}}, x_1)} m(x_0, x_{\text{in}}, x_1) dx_0 dx_{\text{in}} dx_1 =$$

$$\text{KL}\left(m\|\text{proj}_{\mathcal{R}}(m)\right) +$$

$$\int \log \frac{p^{W_\epsilon}(\cancel{x_{\text{in}}|x_0, x_1})[\text{proj}_{\mathcal{R}}(m)](x_0, x_1)}{p^{W_\epsilon}(\cancel{x_{\text{in}}|x_0, x_1})r(x_0, x_1)} m(x_0, x_{\text{in}}, x_1) dx_0 dx_{\text{in}} dx_1 =$$

$$\int \log \frac{[\text{proj}_{\mathcal{R}}(m)](x_0, x_1)}{r(x_0, x_1)} \underbrace{m(x_0, x_1)}_{=[\text{proj}_{\mathcal{R}}(m)](x_0, x_1)} dx_0 dx_1 =$$

$$\text{KL}\left(m\|\text{proj}_{\mathcal{R}}(m)\right) + \int \log \frac{[\text{proj}_{\mathcal{R}}(m)](x_0, x_1)}{r(x_0, x_1)} [\text{proj}_{\mathcal{R}}(m)](x_0, x_1) dx_0 dx_1 =$$

$$\text{KL}\left(m\|\text{proj}_{\mathcal{R}}(m)\right) + \int \log \frac{[\text{proj}_{\mathcal{R}}(m)](x_0, x_1)}{r(x_0, x_1)} [\text{proj}_{\mathcal{R}}(m)](x_0, x_{\text{in}}, x_1) dx_0 dx_{\text{in}} dx_1 =$$

$$\text{KL}\left(m\|\text{proj}_{\mathcal{R}}(m)\right) + \int \log \frac{p^{W_\epsilon}(x_{\text{in}}|x_0, x_1)[\text{proj}_{\mathcal{R}}(m)](x_0, x_1)}{p^{W_\epsilon}(x_{\text{in}}|x_0, x_1)r(x_0, x_1)} [\text{proj}_{\mathcal{R}}(m)](x_0, x_{\text{in}}, x_1) dx_0 dx_{\text{in}} dx_1 =$$

$$\text{KL}\left(m\|\text{proj}_{\mathcal{R}}(m)\right) + \underbrace{\int \log \frac{[\text{proj}_{\mathcal{R}}(m)](x_0, x_{\text{in}}x_1)}{r(x_0, x_{\text{in}}, x_1)} [\text{proj}_{\mathcal{R}}(m)](x_0, x_{\text{in}}, x_1) dx_0 dx_{\text{in}} dx_1}_{=\text{KL}([\text{proj}_{\mathcal{R}}(m)](x_0, x_{\text{in}}x_1)\|r(x_0, x_{\text{in}}, x_1))} =$$

$$= \text{KL}\left(m\|\text{proj}_{\mathcal{R}}(m)\right) + \text{KL}\left(\text{proj}_{\mathcal{R}}(m)\|r\right)$$

That concludes the proof of the second equation (27).

$\square$

**Proposition B.2.** *Assume that we have a sequence of processes $\{q^l\}_{l=0}^\infty$ from D-IMF procedure starting from $q^0$ for which KL $\left(q^0\|q^*\right) < \infty$. Assume that for each reciprocal and Markovian projection in a sequence KL $\left(q^l\|q^{l+1}\right) < \infty$. Then KL $\left(q^{l+1}\|q^*\right) \leq$ KL $\left(q^l\|q^*\right)$ and $\lim_{l\to\infty}$ KL $\left(q^l\|q^{l+1}\right) = 0$.*

*Proof of Proposition B.2.* We use the same technique as was used in the proof of IMF procedure [47, Proposition 7], and for forward KL in [43]. We apply Proposition B.1 and for every $l$ we have:

$$\text{KL}\left(q^l\|q^*\right) = \text{KL}\left(q^l\|q^{l+1}\right) + \text{KL}\left(q^{l+1}\|q^*\right)$$

Since the KL divergence is non-negative, it follows that KL $\left(q^{l+1}\|q^*\right) \leq$ KL $\left(q^l\|q^*\right)$. Applying this proposition for each $l \leq L \in \mathbb{N}$, we have

$$\text{KL}\left(q^0\|q^*\right) = \text{KL}\left(q^0\|q^1\right) + \text{KL}\left(q^1\|q^*\right) = \sum_{l=0}^L \text{KL}\left(q^l\|q^{l+1}\right) + \text{KL}\left(q^{L+1}\|q^*\right).$$

Since KL is non-negative and KL $\left(q^0\|q^*\right) < \infty$, we get $\lim_{l\to\infty}$ KL $\left(q^l\|q^{l+1}\right) = 0$. $\square$

*Proof of Theorem 3.6.* The mild assumptions here are the assumptions of the Propositon B.2, i.e. KL $\left(q^l\|q^{l+1}\right) < \infty$. To prove the current theorem, we follow the proof of [47, Theorem 8] but do the derivations for discrete stochastic processes instead of continuous. By our previous Proposition B.2 it holds that KL $\left(q^l\|q^*\right) \leq$ KL $\left(q^0\|q^*\right) < \infty$ for every $l$. Hence the sequence $(q^l)_{l=0}^\infty$ and its subsequences of markovian $(m^l)_{l=1}^\infty = (q^{2l+1})_{l=1}^\infty$ and reciprocal processes $(r^l)_{l=1}^\infty = (q^{2l})_{l=1}^\infty$ are subsets of a set $\{q \in \mathcal{P}_{2,ac}(\mathbb{R}^{D\times(N+2)}) : \text{KL}(q\|q^*) \leq \text{KL}(q^0\|q^*)\}$ which is compact [50, Theorem 20]. Hence, $(m_l)_{l=1}^\infty$ contains a convergent subsequence $(m^{l_k})_{k=1}^\infty \to m^*$. In turn, the

subsequence $(r^{l_k})_{k=1}^{\infty}$ containes a convergent subsequence $(r^{l_{k_j}})_{j=1}^{\infty} \to r^*$. Since sets of Markovian $\mathcal{M}(N)$ and reciprocal $\mathcal{R}(N)$ processes are closed under weak convergence, we have $m^* \in \mathcal{M}(N)$ and $r^* \in \mathcal{R}(N)$. From the lower semicontinuity of KL divergence in the weak topology [50, Theorem 19] and $\lim_{l \to \infty} \text{KL} \left( q^l \| q^{l+1} \right) = 0$ (see Proposition B.2):

$$0 \leq \text{KL}\left(m^* \| r^*\right) \leq \lim_{j \to \infty} \inf \text{KL}\left(m^{l_{k_j}} \| r^{l_{k_j}}\right) = 0. \tag{28}$$

Thus, $m^* = r^* \overset{\text{def}}{=} q^{\text{lim}}$. We know that $q^{\text{lim}}$ has the same marginals $p_0(x_0) = q(x_0)$ and $p_1(x_1) = q(x_1)$ since both Markovian and reciprocal projections preserve marginals. By our Theorem 3.1 since $q^{\text{lim}} \in \mathcal{M}(N) \cap \mathcal{R}(N)$, then $q^{\text{lim}}(x_0, x_{\text{in}}, x_1) = p^{T^*}(x_0, x_{\text{in}}, x_1)$. Finally, $\lim_{l \to \infty} \text{KL}\left( q^l(x_0, x_{\text{in}}, x_1) \| p^{T^*}(x_0, x_{\text{in}}, x_1) \right) = 0$ follows using

$$\lim_{j \to \infty} \text{KL}\left( r^{l_{k_j}}(x_0, x_{\text{in}}, x_1) \| p^{T^*}(x_0, x_{\text{in}}, x_1) \right) = 0$$

and the mononotonicity of KL $\left( q^l \| q^* \right)$, see Proposition B.2. □

## B.3 Proofs of the Statements in §3.4

The proofs in this subsection are the most technical as there are a lot of manipulations with matrices.

*Proof of Theorem 3.7.* From (6) and (5) follows that the discrete Brownian Bridge $p^{W^\epsilon}(x_{\text{in}}|x_0, x_1)$ has also a Gaussian distribution. The covariance of the Brownian Bridge with coefficient $\epsilon$ at times $s < t$ [17, Eq. 9.14] is $\epsilon s(1 - t)$. Thus, the matrix $\epsilon K$ is a covariance matrix for all pairs of time moments $t, t' \in [t_1, \ldots, t_N]$ of the considered discrete Brownian Bridge $p^{W^\epsilon}(x_{\text{in}}|x_0, x_1)$. The mean value $\mathbb{E}[x_{t_n}|x_0, x_1]$ of Brownian Bridge at time $t_n$ is equal to $t_n x_1 + (1 - t_n)x_0$. Thus, the discrete Brownian Bridge has the following distribution: $p^{W^\epsilon}(x_{\text{in}}|x_0, x_1) = \mathcal{N}(x_{\text{in}}|U x_{01}, \epsilon K)$.

Recall that the reciprocal projection is given by:

$$[\text{proj}_{\mathcal{R}} q](x_{\text{in}}, x_0, x_1) = p^{W^\epsilon}(x_{\text{in}}|x_0, x_1)q(x_0, x_1). \tag{29}$$

Since it is a product of two Gaussian distributions, which itself is also a Gaussian distribution, our goal is to find the mean vector and covariance matrix of $[\text{proj}_{\mathcal{R}} q](x_{\text{in}}, x_0, x_1)$. Further we denote $[\text{proj}_{\mathcal{R}} q](x_{\text{in}}, x_0, x_1)$ as $r(x_0, x_{\text{in}}, x_1)$ for convenience.

The **mean vector** of $[\text{proj}_{\mathcal{R}} q](x_{\text{in}}, x_0, x_1)$ for each $t_n$ is given by

$$\mathbb{E}_{r(x_{t_n})} x_{t_n} = \int \mathbb{E}_{r(x_{t_n}|x_0, x_1)}[x_{t_n}|x_0, x_1]q(x_0, x_1)dx_0 dx_1 =$$

$$\int \mathbb{E}_{p^{W^\epsilon}(x_{t_n}|x_0, x_1)}[x_{t_n}|x_0, x_1]q(x_0, x_1)dx_0 dx_1 = \int \left[ x_0 + t_n(x_1 - x_0) \right] q(x_0, x_1)dx_0 dx_1 =$$

$$(1 - t_n) \int x_0 q(x_0, x_1)dx_0 dx_1 + t_n \int x_1 q(x_0, x_1)dx_0 dx_1 = t_n \mu_1 + (1 - t_n)\mu_0.$$

where $\mu_0$ and $\mu_1$ are the means of $q(x_0)$ and $q(x_1)$, respectively. Thus, the mean vector of $[\text{proj}_{\mathcal{R}} q](x_{\text{in}}, x_0, x_1)$ is given by $(U\mu_{01}, \mu_0, \mu_1)$.

Now, we are going to find the **covariance matrix** $\Sigma_R$. We will first find the inverse covariance

$$\Sigma_R^{-1} = \begin{pmatrix} A & B \\ B^T & C \end{pmatrix}$$

of $[\text{proj}_{\mathcal{R}} q](x_{\text{in}}, x_0, x_1)$. Here $A$ has shape $ND \times ND$ as the matrix $K$, while the matrix $C$ has the shape $2D \times 2D$ as the matrix $\Sigma$. Matrices $A$ and $C$ are symmetric since they are a part of the inversed symmetric matrix $\Sigma_R$. We exploit the fact that the logarithm of a Gaussian distribution has the form (by **Const** we denote all terms that does not depend on $x_{\text{in}}$ or $x_{01}$):

$$\log \left( [\text{proj}_{\mathcal{R}} q](x_{\text{in}}, x_0, x_1) \right) =$$

$$\text{Const} - \frac{1}{2}((x_{\text{in}}, x_{01}) - (U\mu_{01}, \mu_{01}))^T \Sigma_R^{-1}((x_{\text{in}}, x_{01}) - (U\mu_{01}, \mu_{01})) =$$

$$\text{Const} - \frac{1}{2}((x_{\text{in}}, x_{01}) - (U\mu_{01}, \mu_{01}))^T \begin{pmatrix} A & B \\ B^T & C \end{pmatrix} ((x_{\text{in}}, x_{01}) - (U\mu_{01}, \mu_{01})) =$$

$$\text{Const} - \frac{1}{2}(x_{\text{in}} - U\mu_{01})^T A(x_{\text{in}} - U\mu_{01}) - \frac{1}{2}(x_{01} - \mu_{01})^T C(x_{01} - \mu_{01}) -$$

$$(x_{\text{in}} - U\mu_{01})^T B(x_{01} - \mu_{01}) =$$

$$\text{Const} - \frac{1}{2}x_{\text{in}}^T A x_{\text{in}} + (U\mu_{01})^T A x_{\text{in}} - \frac{1}{2}x_{01}^T C x_{01} + \mu_{01}^T C x_{01} -$$

$$x_{\text{in}}^T B x_{01} - x_{\text{in}}^T B \mu_{01} - (U\mu_{01})^T B x_{01} - (U\mu_{01})^T B \mu_{01} =$$

$$\text{Const} - \frac{1}{2}x_{\text{in}}^T A x_{\text{in}} + (U\mu_{01})^T A x_{\text{in}} - \frac{1}{2}x_{01}^T C x_{01} + \mu_{01}^T C x_{01} -$$

$$x_{\text{in}}^T B x_{01} - x_{\text{in}}^T B \mu_{01} - (U\mu_{01})^T B x_{01}.$$

In turn, from (29) we have:

$$\log\left([\text{proj}_{\mathcal{R}} q](x_{\text{in}}, x_0, x_1)\right) = \log p^{W^\epsilon}(x_{\text{in}}|x_0, x_1) + \log q(x_0, x_1) =$$

$$\text{Const} - \frac{1}{2}(x_{\text{in}} - U x_{01})^T (\epsilon K)^{-1}(x_{\text{in}} - U x_{01}) - \frac{1}{2}(x_{01} - \mu_{01})^T \Sigma^{-1}(x_{01} - \mu_{01}) =$$

$$\text{Const} - \frac{1}{2}x_{\text{in}}^T (\epsilon K)^{-1} x_{\text{in}} + x_{\text{in}}^T (\epsilon K)^{-1} U x_{01} - \frac{1}{2}(U x_{01})^T (\epsilon K)^{-1} U x_{01} -$$

$$\frac{1}{2}x_{01}^T \Sigma^{-1} x_{01} + x_{01}^T \Sigma^{-1} \mu_{01} - \frac{1}{2}\mu_{01} \Sigma^{-1} \mu_{01} =$$

$$\text{Const} - \frac{1}{2}x_{\text{in}}^T \underbrace{(\epsilon K)}_{=A} x_{\text{in}} + x_{\text{in}}^T \underbrace{(\epsilon K)^{-1} U}_{=B} x_{01} - \frac{1}{2}x_{01}^T \underbrace{(U^T (\epsilon K)^{-1} U + \Sigma^{-1})}_{=C} x_{01} + x_{01}^T \Sigma^{-1} \mu_{01}.$$

By matching the formulas above, it follows:

$$A = (\epsilon K)^{-1}, \quad B = -(\epsilon K)^{-1} U, \quad C = U^T (\epsilon K)^{-1} U + \Sigma^{-1}. \tag{30}$$

Thus, we have:

$$\Sigma_R^{-1} = \begin{pmatrix} A & B \\ B^T & C \end{pmatrix} = \begin{pmatrix} (\epsilon K)^{-1} & -(\epsilon K)^{-1} U \\ -((\epsilon K)^{-1} U)^T & U^T (\epsilon K)^{-1} U + \Sigma^{-1} \end{pmatrix}$$

By using the formula of block-wise matrix inversion [37, Section 9.1.3] :

$$\begin{pmatrix} A & B \\ B^T & C \end{pmatrix}^{-1} = \begin{pmatrix} A^{-1} + A^{-1} B(C - B^T A^{-1} B)^{-1} B^T A^{-1} & -A^{-1} B(C - B^T A^{-1} B)^{-1} \\ -(C - B^T A^{-1} B)^{-1} B^T A^{-1} & (C - B^T A^{-1} B)^{-1} \end{pmatrix}. \tag{31}$$

Applying this formula, we have:

$$(C - B^T A^{-1} B)^{-1} = (U^T (\epsilon K)^{-1} U + \Sigma^{-1} - U^T (\epsilon K)^{-1} (\epsilon K)(\epsilon K)^{-1} U)^{-1} = (\Sigma^{-1})^{-1} = \Sigma.$$

$$A^{-1} + A^{-1} B(C - B^T A^{-1} B)^{-1} B^T A^{-1} =$$

$$\epsilon K + \epsilon K (\epsilon K)^{-1} U \Sigma \Sigma^{-1} \Sigma U^T \epsilon K (\epsilon K)^{-1} = \epsilon K + U \Sigma U^T.$$

$$-A^{-1} B(C - B^T A^{-1} B)^{-1} = \epsilon K (\epsilon K)^{-1} U \Sigma = U \Sigma.$$

Thus, we obtain the desired result:

$$\Sigma_R = \begin{pmatrix} \epsilon K + U \Sigma U^T & U \Sigma \\ (U \Sigma)^T & \Sigma \end{pmatrix}.$$

$\square$

*Proof of Theorem 3.8.* Part 1. Since from the assumptions of the theorem $q(x_{\text{in}}, x_0, x_1)$ has Gaussian distribution, it follows that joint distribution of two time moments $q(x_{t_n}, x_{t_{n-1}})$ is also Gaussian and is given by:

$$q(x_{t_n}, x_{t_{n-1}}) = \mathcal{N}\left(\begin{pmatrix} x_{t_n} \\ x_{t_{n-1}} \end{pmatrix} \middle| \begin{pmatrix} \mu_{t_n} \\ \mu_{t_{n-1}} \end{pmatrix} \begin{pmatrix} (\widetilde{\Sigma}_R)_{t_n, t_n} & (\widetilde{\Sigma}_R)_{t_n, t_{n-1}} \\ (\widetilde{\Sigma}_R)_{t_{n-1}, t_n} & (\widetilde{\Sigma}_R)_{t_{n-1}, t_{n-1}} \end{pmatrix}\right) \tag{32}$$

Recall that here $(\widetilde{\Sigma}_R)_{t_i, t_j}$ represents submatrix of $\widetilde{\Sigma}_R$ with covariance of $x_{t_i}$ and $x_{t_j}$. Hence, the conditional distributions are given by [37, Sec 8.1.3]:

$$q(x_{t_n}|x_{t_{n-1}}) = \mathcal{N}(x_{t_n}|\widehat{\mu}_{t_n}(x_{t_{n-1}}), \widehat{\Sigma}_{t_n}),$$

$$\widehat{\mu}_{t_n}(x_{t_{n-1}}) \overset{\text{def}}{=} \mu_{t_n} + (\widetilde{\Sigma}_R)_{t_n, t_{n-1}}((\widetilde{\Sigma}_R)_{t_{n-1}, t_{n-1}})^{-1}(x_{t_{n-1}} - \mu_{t_{n-1}}), \tag{33}$$

$$\widehat{\Sigma}_{t_n} \overset{\text{def}}{=} (\widetilde{\Sigma}_R)_{t_n, t_n} - (\widetilde{\Sigma}_R)_{t_n, t_{n-1}}((\widetilde{\Sigma}_R)_{t_{n-1}, t_{n-1}})^{-1}((\widetilde{\Sigma}_R)_{t_n, t_{n-1}})^T.$$

That concludes the first part of our proof about the whole distribution $[\text{proj}_{\mathcal{M}}q](x_0, x_{\text{in}}, x_1)$ of Markovian projection.

Part 2. Next, we find the distribution of $[\text{proj}_{\mathcal{M}}q](x_0, x_1)$, but before we proceed, we introduce new notation to improve readability:

$$q_{\mathcal{M}}(x_0, x_{\text{in}}, x_1) \overset{\text{def}}{=} [\text{proj}_{\mathcal{M}}q](x_0, x_{\text{in}}, x_1). \tag{34}$$

Since the process $q_{\mathcal{M}}(x_0, x_{\text{in}}, x_1)$ is Gaussian, all its joint and conditional distributions are also Gaussian. Moreover, we know that from the definition of the Markovian projection (10) follows that it preserve all marginal distributions, i.e. $q_{\mathcal{M}}(x_{t_n}) = q(x_{t_n})$, hence we can already write that $q_{\mathcal{M}}(x_0, x_1)$ is given by:

$$q_{\mathcal{M}}(x_0, x_1) = \mathcal{N}(\begin{pmatrix} x_0 \\ x_1 \end{pmatrix} | \begin{pmatrix} \mu_0 \\ \mu_1 \end{pmatrix}, \begin{pmatrix} \Sigma_0 & \Sigma_{01} \\ (\Sigma_{01})^T & \Sigma_1 \end{pmatrix}), \tag{35}$$

where $\mu_0$ and $\mu_1$ are the means of $q(x_0)$ and $q(x_1)$, while $\Sigma_0$ and $\Sigma_1$ are the covariance matricies of $q(x_0)$ and $q(x_1)$. Thus, only $\Sigma_{01}$ is unknown. Again, by using the formula for the conditional distributions [37, Sec 8.1.3] we have that:

$$q_{\mathcal{M}}(x_1|x_0) = \mathcal{N}(x_1|\widetilde{\mu}_1(x_0), \widetilde{\Sigma}_1(x_0)),$$

$$\widetilde{\mu}_1(x_0) \overset{\text{def}}{=} \mu_1 + \underbrace{\Sigma_{01}^T \Sigma_0^{-1}}_{\overset{\text{def}}{=} G}(x_0 - \mu_0),$$

$$\widehat{\Sigma}_1 \overset{\text{def}}{=} \Sigma_1 - \Sigma_{01}^T \Sigma_0^{-1} \Sigma_{01}.$$

Since the mean $\widetilde{\mu}_1(x_0)$ of the conditional distribution is a affine map of $x_0$ with the matrix $G$ we can derive:

$$\Sigma_{01}^T = G\Sigma_0.$$

Thus, we need to find the expression for $G$, by considering the expression for $\widetilde{\mu}_1(x_0)$. To derive the expression of the mean $\widetilde{\mu}_1(x_0)$ of $q_{\mathcal{M}}(x_1|x_0)$ we consider the sequence $q_{\mathcal{M}}(x_{t_n}|x_0)$ for $n \in [1, \ldots, N+1]$. We already know the expression for $n = 1$ which is given by $[\text{proj}_{\mathcal{M}}q](x_{t_1}|x_0) = q(x_{t_1}|x_0)$ in the first part of the proof. For other $n$, we use the following expression:

$$q_{\mathcal{M}}(x_{t_n}|x_0) = \int q(x_{t_n}|x_{t_{n-1}})q_{\mathcal{M}}(x_{t_{n-1}}|x_0)dx_{t_{n-1}}. \tag{36}$$

Since $q_{\mathcal{M}}(x_{t_n}|x_0)$ is Gaussian we denote $q_{\mathcal{M}}(x_{t_n}|x_0) = \mathcal{N}(x_{t_n}|\widetilde{\mu}_{t_n}(x_0), \widetilde{\Sigma}_{t_n})$. We derive the mean $\widetilde{\mu}_{t_n}(x_0)$ by using the properties of conditional expectations as follows:

$$\widetilde{\mu}_{t_n}(x_0) = \mathbb{E}_{q_{\mathcal{M}}(x_{t_n}|x_0)}[x_{t_n}] = \int \underbrace{\left(\mathbb{E}_{q(x_{t_n}|x_{t_{n-1}})}[x_{t_n}]\right)}_{\widehat{\mu}_{t_n}(x_{t_{n-1}})} q_{\mathcal{M}}(x_{t_{n-1}}|x_0)dx_{t_{n-1}} =$$

$$\int \left(\mu_{t_n} + (\widetilde{\Sigma}_R)_{t_n, t_{n-1}}((\widetilde{\Sigma}_R)_{t_{n-1}, t_{n-1}})^{-1}(x_{t_{n-1}} - \mu_{t_{n-1}})\right) q_{\mathcal{M}}(x_{t_{n-1}}|x_0)dx_{t_{n-1}} =$$

$$\mu_{t_n} + (\widetilde{\Sigma}_R)_{t_n, t_{n-1}}((\widetilde{\Sigma}_R)_{t_{n-1}, t_{n-1}})^{-1}\left(\left(\int x_{t_{n-1}}q_{\mathcal{M}}(x_{t_{n-1}}|x_0)dx_{t_{n-1}}\right) - \mu_{t_{n-1}}\right) =$$

$$\mu_{t_n} + (\widetilde{\Sigma}_R)_{t_n, t_{n-1}}((\widetilde{\Sigma}_R)_{t_{n-1}, t_{n-1}})^{-1}\left(\left(\underbrace{\mathbb{E}_{q_{\mathcal{M}}(x_{t_{n-1}}|x_0)}x_{t_{n-1}}}_{=\widetilde{\mu}(x_{t_{n-1}})(x_0)}\right) - \mu_{t_{n-1}}\right) =$$

$$\mu_{t_n} + (\widetilde{\Sigma}_R)_{t_n, t_{n-1}}((\widetilde{\Sigma}_R)_{t_{n-1}, t_{n-1}})^{-1}(\widetilde{\mu}_{t_{n-1}}(x_0) - \mu_{t_{n-1}}) = \widehat{\mu}_{t_n}(\widetilde{\mu}_{t_{n-1}}(x_0)). \tag{37}$$

Note that in the line (37), we use equation (33) for $\widehat{\mu}_{t_n}(x_{t_{n-1}})$ with $x_{t_{n-1}} = \widetilde{\mu}_{t_{n-1}}(x_0)$ to simplify the expression. Since $\widetilde{\mu}_{t_n}(x_0) = \widehat{\mu}_{t_n}(\widetilde{\mu}_{t_{n-1}}(x_0))$ we can derive $\widetilde{\mu}_1(x_0)$ recursively as follows:

$$\widetilde{\mu}_1(x_0) = \widetilde{\mu}_{t_{N+1}}(x_0) = \widehat{\mu}_{t_{N+1}}(\widetilde{\mu}_{t_N}(x_0)) = \widehat{\mu}_{t_{N+1}}(\widehat{\mu}_{t_N}(\dots \widehat{\mu}_0(x_0)\dots)),$$

where each $\widehat{\mu}_{t_n}(x_{t_{n-1}})$ is a affine map given by (33) with the matrix given by

$$(\widetilde{\Sigma}_R)_{t_n,t_{n-1}}((\widetilde{\Sigma}_R)_{t_{n-1},t_{n-1}})^{-1}.$$

Hence, $\widetilde{\mu}_1(x_0)$ is a composition of affine maps, and its matrix is given by the product of matrices $\widetilde{\mu}_{t_n}(x_{t_{n-1}})$ as follows:

$$G = \Big[ \prod_{n=1}^{N+1} (\widetilde{\Sigma}_R)_{t_n,t_{n-1}}((\widetilde{\Sigma}_R)_{t_{n-1},t_{n-1}})^{-1} \Big],$$

in turn $\Sigma_{01}^T$ is given by:

$$\Sigma_{01}^T = G\Sigma_0 = \Big[ \prod_{n=1}^{N+1} (\widetilde{\Sigma}_R)_{t_n,t_{n-1}}((\widetilde{\Sigma}_R)_{t_{n-1},t_{n-1}})^{-1} \Big]\Sigma_0.$$

This concludes the proof.

$\square$

## C  Additional Experiments

### C.1  Illustrative 2D Example

Here we consider the SB problem with $p_0$ as a 2D Gaussian distribution and $p_1$ as the Swiss-roll distribution. We use independent $q^0(x_0, x_1) = p_0(x_0)p_1(x_1)$, $N = 3$ ($t_n = \frac{n}{N+1}$) and $K = 20$ outer iterations. We run our ASBM algorithm with different values of parameter $\epsilon$ and present our results in Figure 8. In all the cases, we observe the convergence to the target distribution. Overall, the trajectories are similar to the Brownian bridge and the closeness of start and endpoints is preserved. In Figure 9, we show the evolution of trajectories for different D-IMF iterations, which become more straight when number of iterations increase.

### C.2  Benchmark

We use the SB mixtures benchmark proposed by [13, §4] to experimentally verify that our ASBM algorithm is indeed able to solve the Schrödinger Bridge between $p_0$ and $p_1$. The benchmark provides continuous probability distribution pairs $p_0, p_1$ for dimensions $D \in \{2, 16, 64, 128\}$ with the known static SB solution $p^{T^*}(x_0, x_1)$ for parameters $\epsilon \in \{0.1, 1.10\}$. To evaluate the quality of our recovered SB solution, we use cB$\mathbb{W}_2^2$-UVP metric as suggested by the authors [13, §5] and provide results in Table 1. Additionally, we study how our approach learns the target distribution

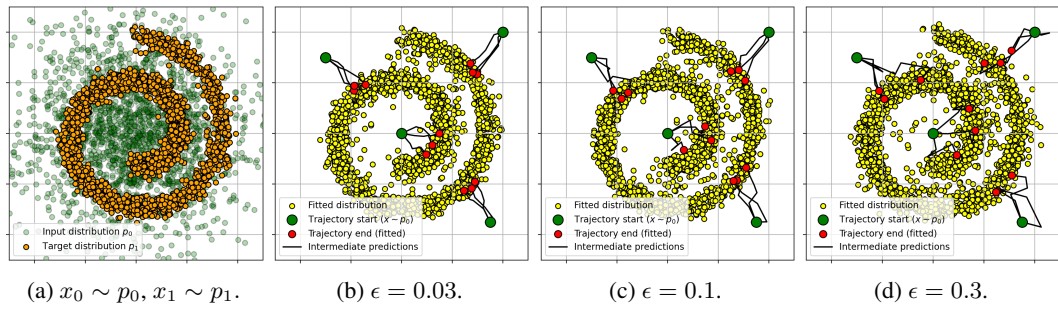

(a) $x_0 \sim p_0, x_1 \sim p_1$.      (b) $\epsilon = 0.03$.      (c) $\epsilon = 0.1$.      (d) $\epsilon = 0.3$.

Figure 8: The final process $q_\theta$ learned with ASBM **(ours)** in *Gaussian→Swiss roll* example.

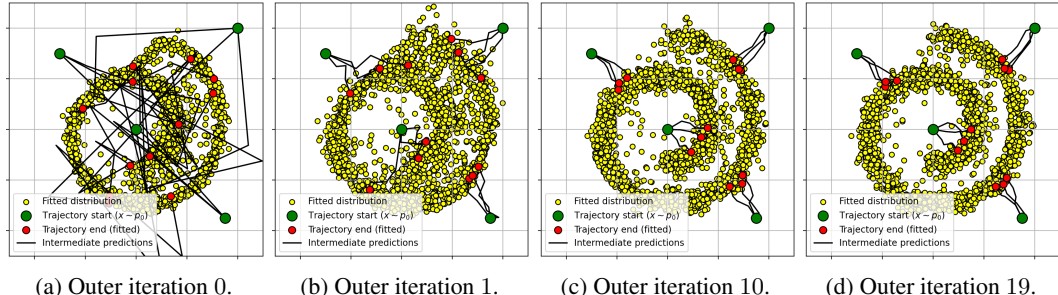

|  (a) Outer iteration 0. | (b) Outer iteration 1. | (c) Outer iteration 10. | (d) Outer iteration 19. |

Figure 9: Evolution of our learned discrete process $q_\theta$ depending on D-IMF iteration in *Gaussian* $\rightarrow$ *Swiss roll* example with $\epsilon = 0.03$.

$p_1$ in Table 2. In all the cases, we run our ASBM algorithm starting from the independent coupling between $p_0$ and $p_1$.

As the baselines, we consider other neural bridge matching methods [49, 47]. The first one (SF$^2$M-Sink) is based on minibatch OT approximations, while the latter implements continuous IMF (DSBM). Additionally, we include the results of the best algorithm (for each setup) from the benchmark [13].

As shown in the Table 1, our algorithm demonstrates superior performance on $\epsilon = 10$, superior performance or comparable performance on $\epsilon = 1$, slightly worse performance w.r.t. SF$^2$M-Sink [49] and superior performance w.r.t. DSBM [47] on $\epsilon = 0.1$. Also, from Table 2 one may note that ASBM fits target distribution better then other Bridge Matching SB algorithms.

| | | $\epsilon = 0.1$ | | | | $\epsilon = 1$ | | | | $\epsilon = 10$ | | | |
| | Algorithm Type | $D=2$ | $D=16$ | $D=64$ | $D=128$ | $D=2$ | $D=16$ | $D=64$ | $D=128$ | $D=2$ | $D=16$ | $D=64$ | $D=128$ |
|---|---|---|---|---|---|---|---|---|---|---|---|---|---|
| Best algorithm on benchmark[†] | Varies | 1.94 | 13.67 | 11.74 | 11.4 | 1.04 | 9.08 | 18.05 | 15.23 | 1.40 | 1.27 | 2.36 | **1.31** |
| DSBM | | 1.21 | 4.61 | 9.81 | 19.8 | 0.68 | **0.63** | **5.8** | 29.5 | 0.23 | 5.45 | 68.9 | 362 |
| SF$^2$M-Sink[†] | Bridge matching | **0.54** | **3.7** | **9.5** | **10.9** | 0.2 | 1.1 | 9 | 23 | 0.31 | 4.9 | 319 | 819 |
| ASBM (**ours**) | | 0.89 | 8.2 | 13.5 | 53.7 | **0.19** | 1.6 | **5.8** | **10.5** | **0.13** | **0.4** | **1.9** | **4.7** |

Table 1: Comparisons of cB$\mathbb{W}_2^2$-UVP $\downarrow$ (%) between the static SB solution $p^T(x_0, x_1)$ and the learned $q_\theta(x_0, x_1)$ on the SB benchmark. The best metric over *bridge Matching algorithms* is **bolded**. Results marked with † are taken from [11].

| | | $\epsilon = 0.1$ | | | | $\epsilon = 1$ | | | | $\epsilon = 10$ | | | |
| | Algorithm Type | $D=2$ | $D=16$ | $D=64$ | $D=128$ | $D=2$ | $D=16$ | $D=64$ | $D=128$ | $D=2$ | $D=16$ | $D=64$ | $D=128$ |
|---|---|---|---|---|---|---|---|---|---|---|---|---|---|
| Best algorithm on benchmark[†] | Varies | 0.016 | 0.05 | 0.25 | 0.22 | 0.005 | 0.09 | 0.56 | 0.12 | 0.01 | 0.02 | 0.15 | 0.23 |
| DSBM | | 0.1 | 0.14 | 0.44 | 3.2 | 0.13 | **0.1** | **0.91** | 6.67 | 0.1 | 5.17 | 66.7 | 356 |
| SF$^2$M-Sink[†] | Bridge matching | 0.04 | 0.18 | **0.39** | **1.1** | 0.07 | 0.3 | 4.5 | 17.7 | 0.17 | 4.7 | 316 | 812 |
| ASBM (**ours**) | | **0.016** | **0.1** | 0.85 | 11.05 | **0.02** | **0.34** | 1.57 | **3.8** | **0.013** | **0.25** | **1.7** | **4.7** |

Table 2: Comparisons of B$\mathbb{W}_2^2$-UVP $\downarrow$ (%) between the ground truth target distribution $p_1(x_1)$ and learned target distribution $q_\theta(x_1)$. The best metric over *bridge matching* algorithms is **bolded**. Results marked with † are taken from [11].

**Remark.** There exist recent light SB algorithms [24, 11] which do not use neural parameterization and rely on the Gaussian mixtures instead. However, these methods have very strong inductive bias towards the benchmark as it is also constructed using Gaussian mixtures. Therefore, we exclude them from comparison, see the comments of the authors in [24, §5.2] and [11, §5.2]

### C.3 Colored MNIST

Here we test ASBM (ours, NFE=4) and DSBM (NFE=100) algorithms starting from mini-batch OT coupling [49] on transfer between colorized MNIST digits of classes "2" and "3" with $\epsilon \in \{1, 10\}$. We learn ASBM and DSBM on *train* set of digits and show the translated *test* images in Figures 10 and 11 along with calcualted *test* FID in Table 3.

For $\epsilon = 1$ the color stays almost exactly the same through translation and there are minor shape diversity for both ASBM and DSBM, see Figures (10b, 10c, 11b, 11c). In turn, $\epsilon = 10$ introduces more stochastisity to the solutions, and expectedly the color and shape vary a bit but overall stays similar to input data for both ASBM and DSBM, see Figures (10d, 10e, 11d, 11e). As one can see from Table 3, ASBM has better FID on both $\epsilon \in \{1, 10\}$. However DSBM experiences a notable

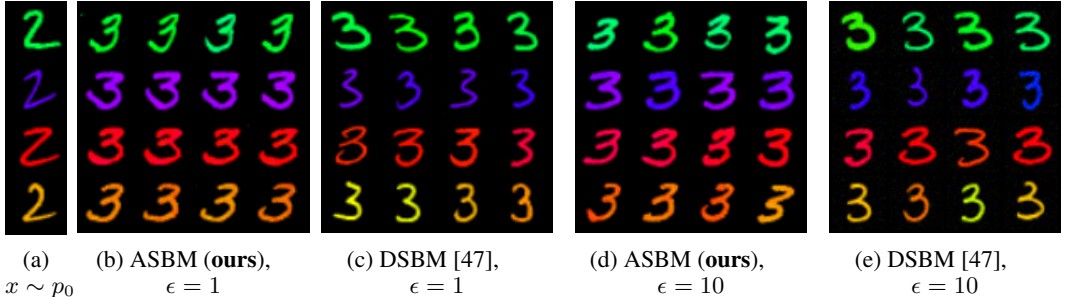

|  (a)  | (b) ASBM (**ours**), | (c) DSBM [47], | (d) ASBM (**ours**), | (e) DSBM [47], |
|:---:|:---:|:---:|:---:|:---:|
| $x \sim p_0$ | $\epsilon = 1$ | $\epsilon = 1$ | $\epsilon = 10$ | $\epsilon = 10$ |

Figure 10: Samples from ASBM (**ours**) and DSBM learned on Colored MNIST *2→3* ($32 \times 32$) translation for $\epsilon \in \{1, 10\}$.

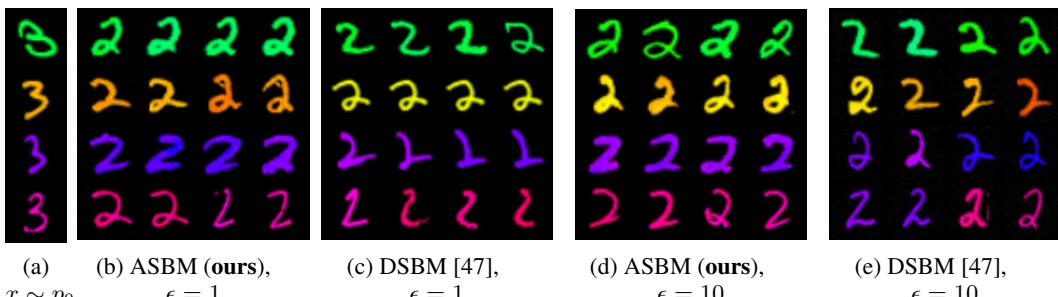

|  (a)  | (b) ASBM (**ours**), | (c) DSBM [47], | (d) ASBM (**ours**), | (e) DSBM [47], |
|:---:|:---:|:---:|:---:|:---:|
| $x \sim p_0$ | $\epsilon = 1$ | $\epsilon = 1$ | $\epsilon = 10$ | $\epsilon = 10$ |

Figure 11: Samples from ASBM (**ours**) and DSBM learned on Colored MNIST *3→2* ($32 \times 32$) translation for $\epsilon \in \{1, 10\}$.

| Model | $\epsilon$ | FID ($2 \to 3$) | FID ($3 \to 2$) |
|:---:|:---:|:---:|:---:|
| ASBM (**ours**) | 1 | 2.7 | 2.8 |
| DSBM | 1 | 6.2 | 5.3 |
| ASBM (**ours**) | 10 | 4.3 | 4.53 |
| DSBM | 10 | 58.7 | 59.9 |

Table 3: C-MNIST FID $\downarrow$ values for ASBM and DSBM with $\epsilon \in \{1, 10\}$

increase in FID with $\epsilon = 10$. We conjecture that this is due to the FID instability w.r.t. slightly noisy images which may appear in DSBM due to the neccesity to integrate noisy trajectories (for large $\epsilon$).

## D    Experimental Details

### D.1    Details of DDGAN Implementation for Learning Markovian Projection

Below, we discuss the parametrization of the discriminator and generator in detail. In general we follow [53], but we change their DDPM diffusion inner process on the Brownian bridge process.

**Parametrization and objective for the discriminator.** As in the DD-GAN paper [53] we use a time-conditional discriminator $D_\xi(x_{t_n}, x_{t_{n-1}}, t_{n-1})$: $\mathbb{R}^D \times \mathbb{R}^D \times [0, 1] \to [0, 1]$. For each time moment $t$ and object $x_{t_{n-1}}$, the role of this discriminator is to check whether the sample $x_{t_n}$ is from the distribution $q(x_{t_n}|x_{t_{n-1}})$. As well as in the DD-GAN paper [53], we train this discriminator by optimizing the following objective:

$$\min_\xi \sum_{n=1}^{N+1} \mathbb{E}_{q(x_{t_{n-1}})}[\mathbb{E}_{q(x_{t_n}|x_{t_{n-1}})}[-\log D_\xi(x_t, x_{t_{n-1}}, t_{n-1})] \tag{38}$$

$$+\mathbb{E}_{q_\theta(x_{t_n}|x_{t_{n-1}})}[-\log\big(1 - D_\xi(x_t, x_{t_{n-1}}, t_{n-1})\big)]]$$

Here, the samples from $q(x_{t_n}|x_{t_{n-1}})$ play the role of true samples, while the samples obtained from the parametrized distribution $q_\theta(x_{t_n}|x_{t_{n-1}})$ play role of fake samples in terms of original GANs. To estimate the first expectation $\mathbb{E}_{q(x_{t_{n-1}})}\mathbb{E}_{q(x_{t_n}|x_{t_{n-1}})} = \mathbb{E}_{q(x_{t_n}, x_{t_{n-1}})}$ one should sample from $q(x_{t_n}, x_{t_{n-1}})$. To sample a pair $(x_{t_n}, x_{t_{n-1}}) \sim q(x_{t_n}, x_{t_{n-1}})$, we use the properties (5) and (6) of

the reciprocal process $q$:

$$q(x_{t_n}, x_{t_{n-1}}) = \int p^{W^\epsilon}(x_{t_n}|x_{t_{n-1}}, x_1)p^{W^\epsilon}(x_{t_{n-1}}|x_0, x_1)q(x_1, x_0)dx_1 dx_0.$$

Sampling from $q(x_{t_{n-1}})q_\theta(x_{t_n}|x_{t_{n-1}})$ for estimation of second expectation is given in detail below.

**Parametrization and objective for the generator.** We follow the same setup as the authors of DD-GAN [53] and parametrize $q_\theta(x_{t_n}|x_{t_{n-1}})$ implicitly through the generator $G_\theta(x_{t_{n-1}}, z, t)$ : $\mathbb{R}^D \times \mathbb{R}^Z \times [0, 1] \to \mathbb{R}^D$ as follows:

$$q_\theta(x_{t_n}|x_{t_{n-1}}) \stackrel{\text{def}}{=} \int_{\mathbb{R}^D} q_\theta(x_1|x_{t_{n-1}})p^{W^\epsilon}(x_{t_n}|x_{t_{n-1}}, x_1)dx_1 =$$

$$\int_{\mathbb{R}^Z} p^{W^\epsilon}(x_{t_n}|x_{t_{n-1}}, x_1 = G_\theta(x_{t_{n-1}}, z, t))p_z(z)dz,$$

where $q_\theta(x_1|x_{t_{n-1}})$ should match $q(x_1|x_{t_{n-1}})$ and $p_z(z)$ is the auxiliary probability distribution for the generator $G_\theta$ to model samples from $q_\theta(x_1|x_{t_{n-1}})$. Thus, for a given $x_{t_{n-1}}$ sample $x_{t_n} \sim q_\theta(x_{t_n}|x_{t_{n-1}})$ is obtained by first sampling $x_1$ from the generator $G_\theta$ and then using sampling from the Brownian bridge $p^{W^\epsilon}(x_{t_n}|x_{t_{n-1}}, x_1)$. While in the DD-GAN, the authors use the intermediate time distribution $q(x_{\text{in}}|x_0, x_1)$ from DDPM [15] and it is the main difference between our Markovian projection and one which the authors of DD-GAN used. As in the non-saturation GANs [10], we train the generator by optimizing the following objective:

$$\max_\theta \sum_{n=1}^{N+1} \mathbb{E}_{q(x_{t_{n-1}})}\mathbb{E}_{q_\theta(x_{t_n}|x_{t_{n-1}})}[\log\big(D_\phi(x_t, x_{t_{n-1}}, t_{n-1})\big)].$$

## D.2 Details of D-IMF Implementation

**General description of the ASBM algorithm.** D-IMF algorithm is parametrized by the number $K$ of outer D-IMF iterations, number of inner D-IMF iterations (number of generator gradient optimization steps inside one IMF iteration), ASBM number of inner steps $N$ and starting coupling $q^0(x_0, x_1)$ used in the initial reciprocal process $q^0(x_0, x_{\text{in}}, x_1) = p^{W^\epsilon}(x_{\text{in}}|x_0, x_1)q^0(x_0, x_1)$. Our ASBM Algorithm 1 for D-IMF procedure is analog of DSBM [47, Algorithm 1] for IMF procedure.

---

**Algorithm 1:** Adversarial SB matching (ASBM).

---

**Input** : number of intermediate steps $N$;
        initial process $q^0(x_0, x_{t_1}, \ldots, x_{t_N}, x_1)$ accessible by samples;
        number of outer iteration $K \in \mathbb{N}$;
        forward transitional density network $\{q_\theta(x_{t_n}|x_{t_{n-1}})\}_{n=1}^{N+1}$;
        backward transitional density network $\{q_\eta(x_{t_{n-1}}|x_{t_n})\}_{n=1}^{N+1}$;
**Output** : $p_0(x_0) \prod_{n=1}^{N+1} q_\theta(x_{t_n}|x_{t_{n-1}}) \approx p_1(x_1) \prod_{n=1}^{N+1} q_\eta(x_{t_{n-1}}|x_{t_n}) \approx p^{T^*}(x_0, x_{\text{in}}, x_1)$.
**for** $k = 0$ **to** $K - 1$ **do**
    | Learn $\{q_\theta(x_{t_n}|x_{t_{n-1}})\}_{n=1}^{N+1}$ using 15 with $q^{4k}$;
    | Let $q^{4k+1}$ be given by $p_0(x_0) \prod_{n=1}^{N+1} q_\theta(x_{t_n}|x_{t_{n-1}})$;
    | Let $q^{4k+2}$ be given by $p^{W^\epsilon}(x_{\text{in}}|x_0, x_1)q_\theta(x_0, x_1)$;
    | Learn $\{q_\eta(x_{t_{n-1}}|x_{t_n})\}_{n=1}^{N+1}$ using 16 with $q^{4k+2}$;
    | Let $q^{4k+3}$ be given by $p_1(x_1) \prod_{n=1}^{N+1} q_\eta(x_{t_{n-1}}|x_{t_n})$;
    | Let $q^{4k+4}$ be given by $p^{W^\epsilon}(x_{\text{in}}|x_0, x_1)q_\eta(x_0, x_1)$;

---

We do not reinitialize neural networks during the ASBM algorithm.

**Special pretraining on the $0$-th outer iteration.** While, in general, Algorithm 1 implements our scheme, in our experiments, we slightly modify the initial outer iteration based on purely empirical reasons. We train both forward and backward models $\{q_\theta(x_{t_n}|x_{t_{n-1}})\}_{n=1}^{N+1}$ and $\{q_\eta(x_{t_{n-1}}|x_{t_n})\}_{n=1}^{N+1}$ with $q^0$ and the let $q^1$ be $p_0(x_0) \prod_{n=1}^{N+1} q_\theta(x_{t_n}|x_{t_{n-1}})$. We use more gradient setups on this iteration than on the further outer iterations. We do that to "pretrain" both processes $q_\theta$ and $q_\eta$ to model $p_1$ and $p_0$ respectively. Then we proceed to other iterations as described in Algorithm 1.

## D.3 Hyperparameters of ASBM

For all the experiments, Discrete Markovian Projection is conducted using the DD-GAN code [53]:

```
https://github.com/NVlabs/denoising-diffusion-gan
```

The only thing that we modify is the replacement of the DDPM [15] posterior sampling for generator with our Brownian Bridge posterior sampling, see Appendix D.1. In all the experiments we use a uniform time discretization, i.e., for the number of inner times points $N$, $t_n = \frac{n}{N+1}$ for $n \in [0, N+1]$.

In Toy 2D (Appendix C.1) and SB Benchmark (Appendix C.2) experiments, both generator and discrimintor are parametrized by MLPs with inner layer widths $[256, 256, 256]$, LeakyReLU activations and 2-dimensional time embeddings using `torch.nn.Embeddings`. In CelebA (§4.2) and Colored MNIST (Appendix C.3) experiments, generator is parametrized by U-Net [42] and discriminator by a ResNet-like architectures with addition of positional time encoding as in [53]. Neural networks are optimized with the Adam optimizer [22] and apply the Exponential Moving Averaging (EMA) on generator's weights. At the start of a new D-IMF iteration, both the generator, generator (EMA), discriminator and optimizers are initialized using checkpoints from the end of the previous D-IMF iteration. Inside each D-IMF iteration (except the initial one), EMA generator weights are used for sampling from previous Discrete Markovian Projections. Starting coupling $q^0(x_0, x_1)$ may be either Ind, i.e. $q^0(x_0, x_1) = p_0(x_0)p_1(x_1)$, or Mini Batch Optimal Transport coupling (MB), i.e. discrete Optimal Transport solved on mini-batch samples [49].

The hyperparameters which we use in the experiments are summarized in Table 4.

| Experiment | Start couping $q^0(x_0, x_1)$ | D-IMF outer iters | D-IMF=0 grad updates | D-IMF grad updates | $N$ | Batch Size | $D/G$ opt ratio | EMA decay | Lr $G$ | Lr $D$ |
|---|---|---|---|---|---|---|---|---|---|---|
| 2D Toy | Ind | 20 | 400000 | 40000 | 3 | 512 | 1:1 | 0.999 | 1e-4 | 1e-4 |
| SB Bench | Ind | 2 | 133000 | 67000 | 31 | 128 | 3:1 | 0.999 | 1e-4 | 1e-4 |
| C-MNIST | MB | 3 | 100000 | 50000 | 3 | 64 | 1:1 | 0.999 | 1.25e-4 | 1.6e-4 |
| CelebA | MB | 5 | 1000000 | 40000 | 3 | 32 | 1:1 | 0.9999 | 1.25e-4 | 1.6e-4 |

Table 4: Hyperparameters for experiments. $D$ stands for Discriminator and $G$ stands for Generator. Ratio of Discriminator optimization steps w.r.t. Generator optimization steps is denoted by $D/G$ opt ratio. Lr stands for learning rate.

**Other details & pre-processing.** Test FID is calculated using pytorch-fid package. Working with CelabA dataset [33], we use all 84434 male and 118165 female samples (90% train, 10% test of each class). Each sample is resized to $128 \times 128$ and normalized by 0.5 mean and 0.5 std. Generator and discriminator are the same as for CelebA-HQ in DDGAN [53] (42M Generator parameters and 27M Discriminator parameters). Working with Colorized MNIST [12], we pick digits of classes "2" and "3" (we use the default MNIST *train/test* split), resize them to $32 \times 32$ and normalize by 0.5 mean and 0.5 std. We use the same generator and discriminator as DDGAN uses in CIFAR10 [53].

**Computational time.** The most time challenging experiment on CelebA runs for approximately 7 days on 1 GPUs A100. Experiment with Colored MNIST takes less then 2 days of training on GPU A100. Toy2D and Schrödinger Bridge benchmark experiments take several hours on GPU A100.

## D.4 Details of DSBM Baseline

DSBM [47] implementation is taken from the official code:

```
https://github.com/yuyang-shi/dsbm-pytorch
```

For CelebA experiment all the hyperparameters, except for 200k training iterations for the first IMF iteration (Bridge Matching **pretrain**, Appx I.3 [47]) and number of overall IMF iterations (that is taken the same as for corresponding ASBM experiment, see Table 4), were taken from [47]. As a neural network time conditional U-Net model (38M parameters) was used. Hyperparameters and neural network for Colored MNIST experiment were taken from MNIST $\leftrightarrow$ E-MNIST experiment [47, §6]. Starting coupling is exactly the same as for ASBM in corresponding experiments (Table 4).

# E  Additional results on CelebA

## E.1  Extended Evaluation using Other Metrics

**FID for *female→male*.**  We evaluate the backward model (*female→male*) trained for unpaired CelebA (128×128) image-to-image translation (§4.2) and present the test FID in Table 5.

| Model | $\epsilon = 1$ | $\epsilon = 10$ |
|---|---|---|
| DSBM | 24.06 | 92.15 |
| ASBM (ours) | 16.86 | 17.44 |

Table 5: Test FID↓ values for CelebA *female→male* image-to-image translation.

**CMMD.**  To strengthen the unpaired CelebA (128×128) *male→female* image-to-image translation (§4.2) experimental results, we add CMMD [19] metric. CMMD is a recent analogue of the FID that enjoys unbiased estimation and rich CLIP [40] embeddings. We estimate CMMD on CelebA for the same DSBM and ASBM models as for the FID calculation (§7) using all available *female* test samples and present results in Table 6. It can be seen that the CMMD values correlate with the FID values.

| Model | $\epsilon = 1$ | $\epsilon = 10$ |
|---|---|---|
| DSBM | 0.365 | 1.140 |
| ASBM (ours) | 0.216 | 0.231 |

Table 6: CMMD↓ [19] metric for unpaired CelebA (128×128) *male→female* image-to-image translation estimated on *female* test set.

**Training with different NFE.**  In the unpaired CelebA (128×128) *male→female* image-to-image translation (§ 4.2), number of inner steps $N = 3$ is considered. However, it is possible to train the model with different values of $N$, which correspond to the model NFE minus one. For completeness, we provide experimental results with training and evaluation at $N = 1$ and $N = 7$ (NFE= 2 and NFE= 8). Here all training hyperparameters are the same as for $N = 3$, see Appendix D. Samples and test FID are shown in Figure 12.

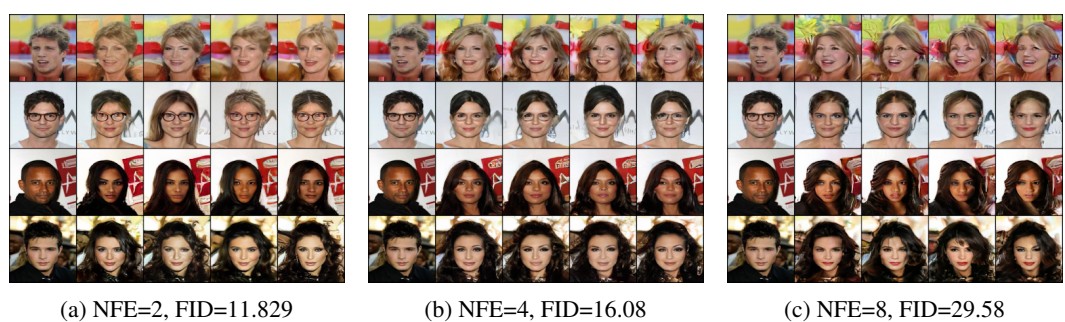

|               (a) NFE=2, FID=11.829               |               (b) NFE=4, FID=16.08               |               (c) NFE=8, FID=29.58               |

Figure 12: Unpaired CelebA (128×128) *male→female* image-to-image translation samples and FID↓ values for ASBM trained with different NFE $\in \{2, 4, 8\}$ with $\epsilon = 1$.

**Inference with different NFE.**  Although in practice models are trained with a fixed NFE (see Appendix D), it is possible to use different NFE at the inference stage by exploiting the continuity of the time-conditional module, see Algorithm 2. We take the model for *male→female* trained on NFE=3 with $\epsilon = 1$ and evaluate it with different NFE $\in \{1, 2, 3, 4, 8, 16, 32\}$, see the results in Figure 13, and do quantitative evaluation using FID and MSE cost (MSE between inputs and outputs) in Table 7. As can be seen, the MSE cost increases with NFE and the FID is optimal at NFE=4.

**Algorithm 2:** Inference of forward ASBM model.

**Input**  : number of intermediate steps $N$; sample $x_0$
forward $x_N$ generator network $G_\theta$;

**Output** : sample from $p_0(x_0) \prod_{n=1}^{N+1} q_\theta(x_{t_n}|x_{t_{n-1}})$.

**for** $n = 0$ **to** $N$ **do**
   $x_{n+1} \sim p^{W^\epsilon}(x_{t_{n+1}}|x_{t_n}, x_1 = G_\theta(x_{t_n}, z, t))$

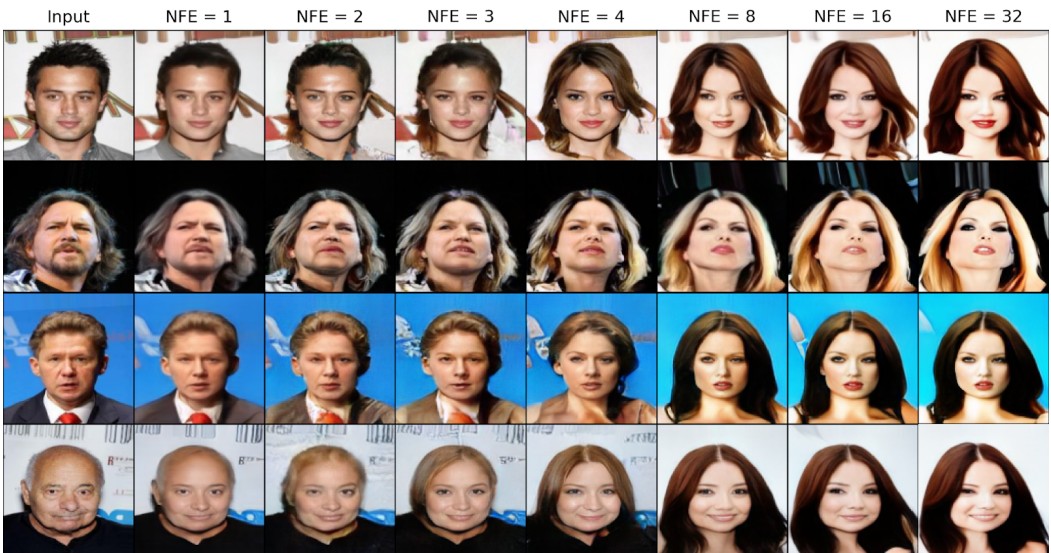

Input   NFE = 1   NFE = 2   NFE = 3   NFE = 4   NFE = 8   NFE = 16   NFE = 32

Figure 13: CelebA *male→female* translation samples of ASBM trained with NFE$= 4$ and evaluated with NFE $\in \{1, 2, 3, 4, 8, 16, 32\}$.

| NFE | 1 | 2 | 3 | 4 | 8 | 16 | 32 |
|---|---|---|---|---|---|---|---|
| FID | 58.71 | 32.27 | 17.67 | 16.62 | 55.72 | 67 | 86.97 |
| MSE cost | 0.009 | 0.023 | 0.047 | 0.113 | 0.288 | 0.354 | 0.50 |

Table 7: Quantitative evaluation of ASBM model trained with NFE$= 4$ and evaluated with NFE$\in \{1, 2, 3, 4, 8, 16, 32\}$. FID↓ and MSE cost are calculated on the test set.

**LPIPS diversity**. To measure the generation diversity of our model on CelebA *male→female* translation, we compute the LPIPS variance [16]. Specifically, we take a subset of 500 images from the test part of the Celeba dataset and sample a batch of 16 generated images for each input image. We then compute the average LPIPS [55] distance between all possible pairs of these images and average these values. We present the results in the Table 8 for DSBM and ASBM with different values of the coefficient $\epsilon = 1$ and $\epsilon = 10$.

| Model | $\epsilon = 1$ | $\epsilon = 10$ |
|---|---|---|
| DSBM | 0.1047 | 0.1909 |
| ASBM (ours) | 0.0933 | 0.1878 |

Table 8: Average diversity of DSBM and ASBM generative models for *male→female* translation measured by using LPIPS variance [16].

**LPIPS perceptual similarity**. To evaluate the content preservation during the unpaired image-to-image *male→female* translation on CelebA, we calculate the perceptual similarity. Namely, we take the test samples from CelebA dataset, translate them using learned DSBM and ASBM models with parameters $\epsilon = 1$ and $\epsilon = 10$ and then calculate LPIPS [55] between inputs and generated outputs and average results. One can see results in the Table 9.

| Model | $\epsilon = 1$ | $\epsilon = 10$ |
|-------|------|-------|
| DSBM | 0.246 | 0.386 |
| ASBM | 0.242 | 0.294 |

Table 9: Perceptual similarity for *male→female* translation for DSBM and ASBM models with $\epsilon = 1$ and $\epsilon = 10$ measured using LPIPS↓ [55] between inputs from CelebA test and generated outputs.

## E.2 Analysis on D-IMF/IMF iterations dynamics

We include additional analysis on dynamics of model samples with D-IMF iterations for ASBM (ours) and IMF iterations for DSBM with $\epsilon = 1$. As one can see from Figure 15a ASBM visually almost converges after 5 iterations in terms of similarity of generated sample w.r.t. to input data, i.e., the transport cost. From plot in Figure 14 we see that ASBM's FID does not change through subsequent D-IMF iterations; ASBM fits target on the iteration 5 rather well. Looking at Figure 15b, one can conclude that for DSBM visual similarity along side with transport cost starts to diverge after 5th outer IMF iteration. Also, as it can be seen at plot in Figure 14, FID stops to improve after outer iteration 9 and does not improve drastically

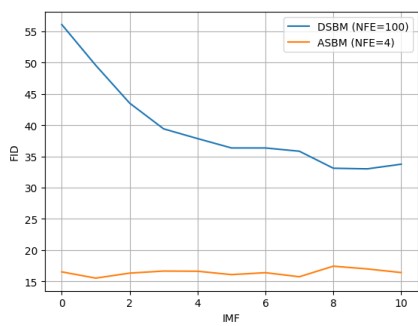

Figure 14: ASBM and DSBM FID w.r.t. IMF iterations.

from outer iteration 5. Hence, we take ASBM and DSBM with 5 outer D-IMF/IMF iterations as a balance point for our comparison.

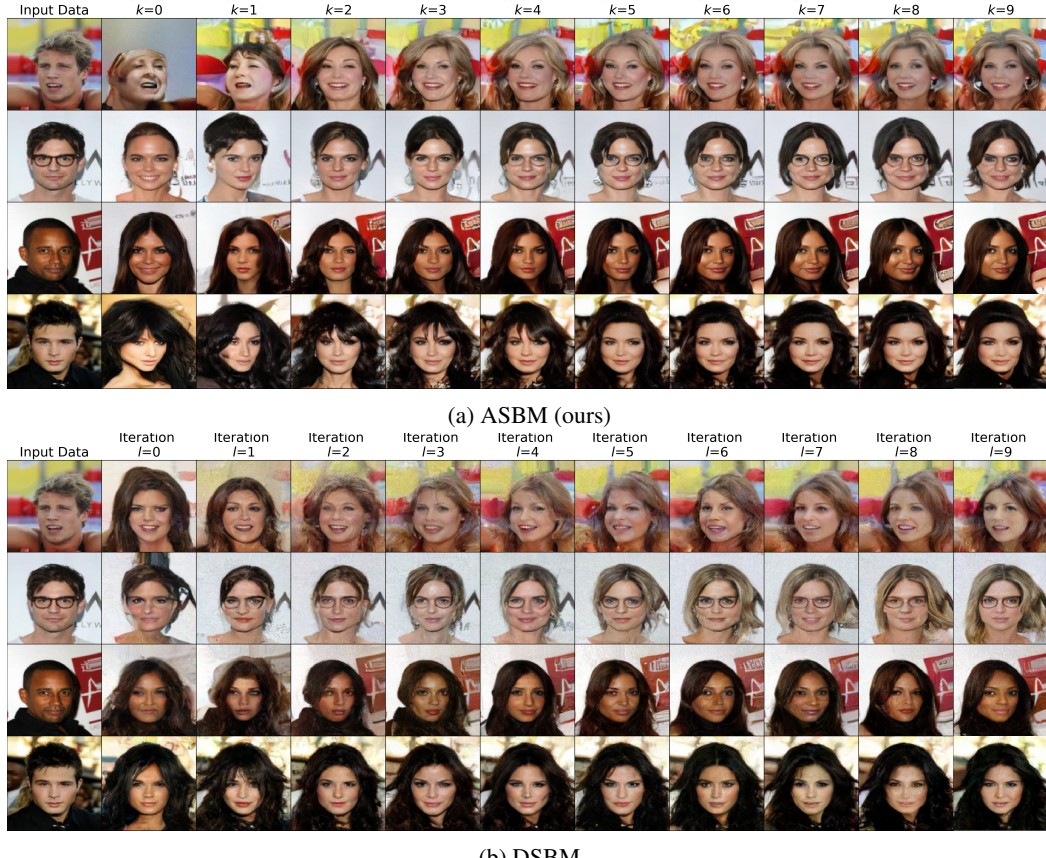

(a) ASBM (ours)

(b) DSBM

Figure 15: Samples dependence on D-IMF/IMF outer iterations number $k$, $\epsilon = 1$.

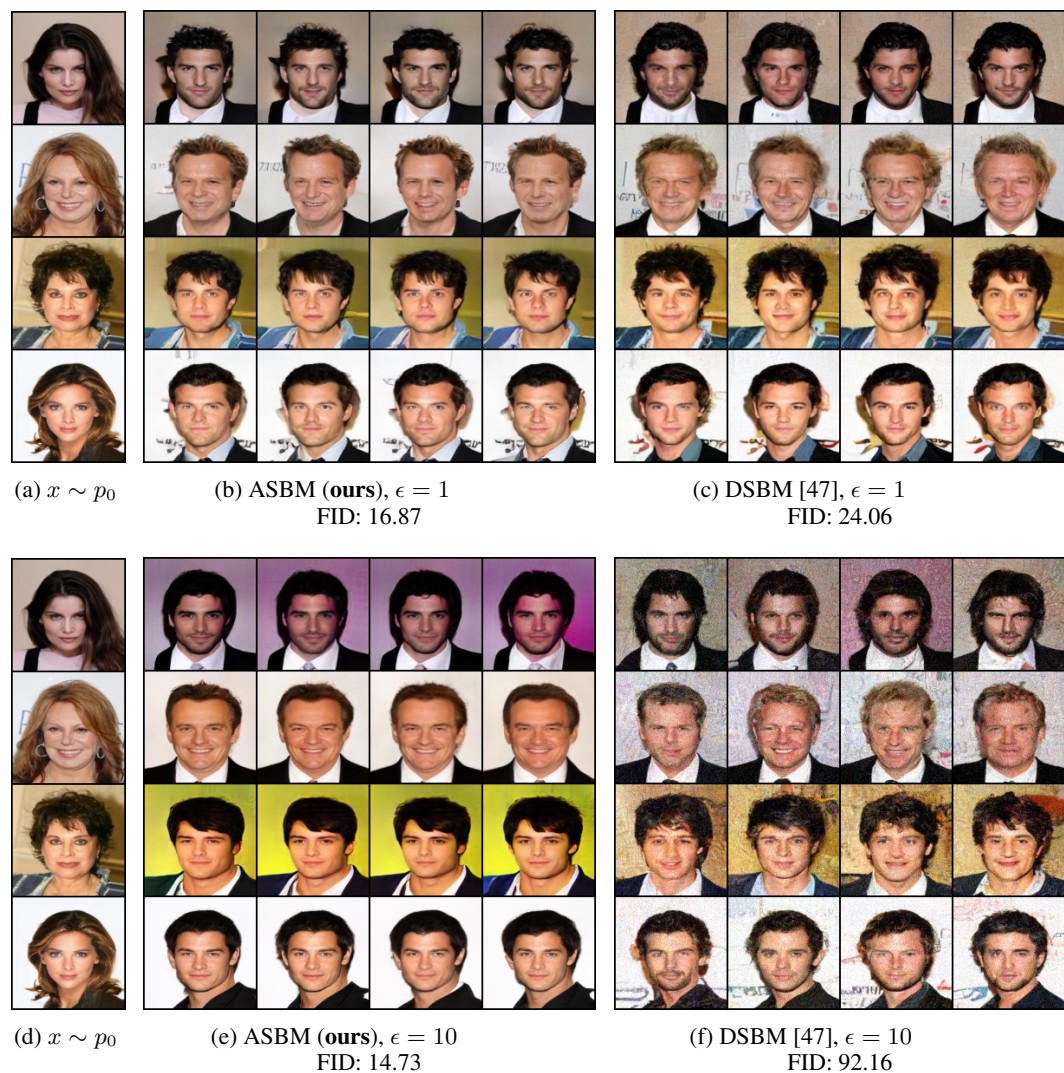

(a) $x \sim p_0$      (b) ASBM (**ours**), $\epsilon = 1$      (c) DSBM [47], $\epsilon = 1$
FID: 16.87      FID: 24.06

(d) $x \sim p_0$      (e) ASBM (**ours**), $\epsilon = 10$      (f) DSBM [47], $\epsilon = 10$
FID: 14.73      FID: 92.16

Figure 16: Samples from ASBM (ours) and DSBM learned on Celeba *female→male* ($128 \times 128$) for $\epsilon \in \{1, 10\}$

### E.3 ASBM (ours) and DSBM samples for *female→male* ($128 \times 128$)

In Figure 16, we provide additional examples for *female→male* ($128 \times 128$) setting with $\epsilon \in \{1, 10\}$ for ASBM (Figures 16b, 16e) and DSBM (Figures 16c, 16f) along with quantitative evaluation of FID values. Both ASBM and DSBM models were evaluated at D-IMF/IMF iteration number 4. As one can see ASBM (NFE=4) outperforms DSBM (NFE=100) in FID using only 4 evaluation steps.

### E.4 Extra (uncurated) samples for ASBM (ours) on CelebA *male↔female* ($128 \times 128$)

In Figures 17 and 18, we provide additional samples for ASBM CelebA *male↔female* ($128 \times 128$) experiment with $\epsilon \in \{1, 10\}$.

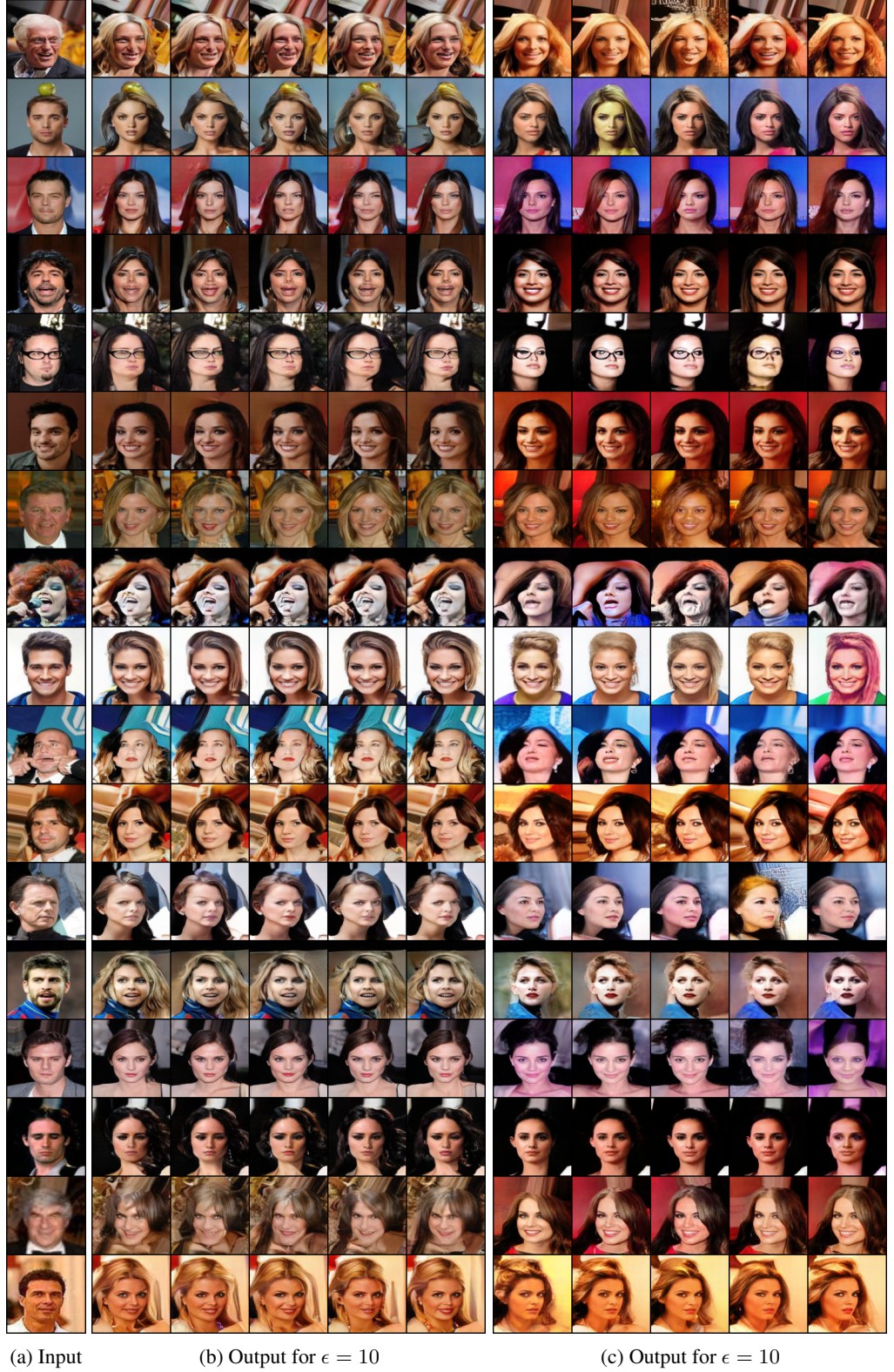

(a) Input        (b) Output for $\epsilon = 10$        (c) Output for $\epsilon = 10$

Figure 17: ASBM (ours) Celeba *male→female* ($128 \times 128$) samples for $\epsilon \in \{1, 10\}$
.

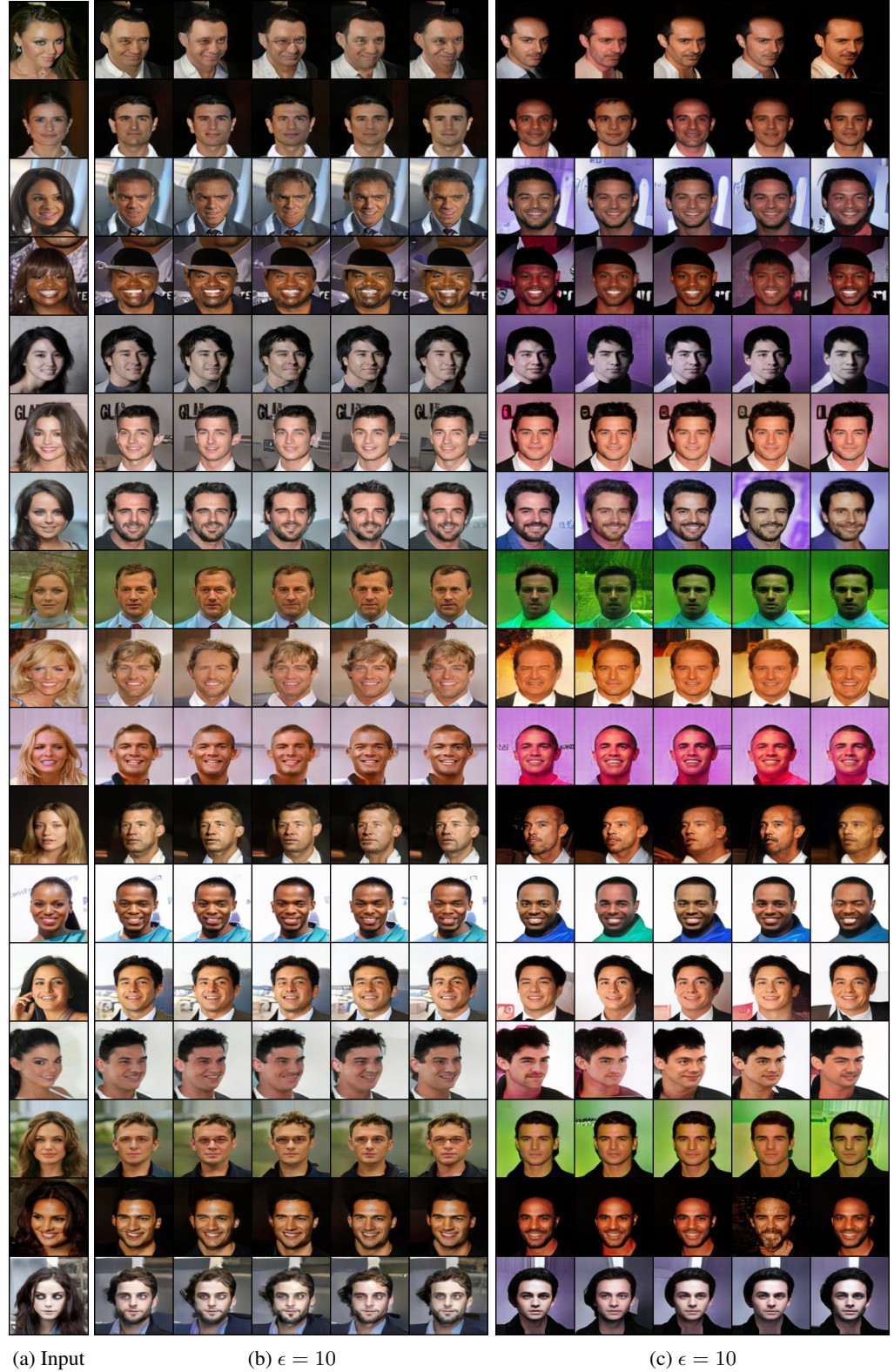

(a) Input             (b) $\epsilon = 10$             (c) $\epsilon = 10$

Figure 18: ASBM *female→male* ($128 \times 128$) samples for $\epsilon \in \{1, 10\}$

