# OpenReview forum: "Adversarial Schrödinger Bridge Matching"
_NeurIPS.cc/2024/Conference — NeurIPS 2024 poster_

### Official Review · Reviewer_LCNa · 2024-06-21

**Soundness:** 3
**Presentation:** 3
**Contribution:** 3
**Rating:** 6
**Confidence:** 1

**Summary:**

This paper proposes Discrete-time iterative Markovian Fitting, which is an efficient alternative that greatly reduces NFE from IMF (Shi et al. 2023). This was possible due to the tractability of Brownian bridges for any subintervals, so discretization of IMF does not hurt the performance of if the model can be trained on it. For training a generative model, the authors adopted a time-dependent GAN called Denoising Diffusion Generative Adversarial Models (DD-GAN). Compared to DSBM, the proposed ASBM shows significantly low computational costs with good FID scores.

**Strengths:**

Overall, I think this paper is well written and quality of presenting idea and theoretical statements are very good. This work also contains certain contributions for probabilistic generative models.
- This work broadens our understanding of IMF, reference measures, and the application of Brownian bridges.
- The results verified that GANs hold as a good nonlinear probabilistic generative model training method.
- Theoretical statements were clear and completely proved.

**Weaknesses:**

- I strongly believe the performance gains are mostly from applying GAN architecture, since D-IMF and IMF indicates essentially the same learning scheme.  Therefore, it is indeed an improvement from DSBM, but novelty of this work as a distinct SB study might not be significant from the perspective of diffusion models.
- This model does not always yield state-of-the-art scores for syntactic datasets (see Table 1&2).
- While I understand this work for some extent, I do not clearly understand the motivation why do we need a sampling-based approach when we have sampling-free method such as SF^2M and LightSB [1] models

Typo
- L213: Denoising Denoising GANs

[1]
@inproceedings{
korotin2024light,
title={Light Schr\"odinger Bridge},
author={Alexander Korotin and Nikita Gushchin and Evgeny Burnaev},
booktitle={The Twelfth International Conference on Learning Representations},
year={2024},
url={https://openreview.net/forum?id=WhZoCLRWYJ}
}

**Questions:**

How do the authors suspect that the FID score is very low for ASBM? Following the arguments of [1], could the authors report CMMD scores or other metrics for results in Fig. 3?

[1] https://arxiv.org/pdf/2401.09603

**Limitations:**

The authors adequately addressed the limitations

---

> ### Author Rebuttal · Authors · 2024-08-07
>
> Dear Reviewer LCNa, thank you for your comments. Here are the answers to your questions and comments.
>
> **(1) I strongly believe the performance gains are mostly from applying GAN architecture, since D-IMF and IMF indicates essentially the same learning scheme. Therefore, it is indeed an improvement from DSBM, but the novelty of this work as a distinct SB study might not be significant from the perspective of diffusion models.**
>
> It is not possible to just apply GAN architecture in the original continuous IMF framework [2] since their Markovization objective is based on bridge-matching and implies learning a drift function of SDE and not transitional densities of a discrete process. The main cause of it is that the continuous IMF framework is based on the known property that SB is a unique **continuous** Markovian and reciprocal process.
>
> The novelty of our work as a distinct SB study comes from our main theorem (Theorem 3.1). We show that to solve the SB problem, it is enough to find a probability distribution $\pi(x_0, x_{t_{1}}, \dots, x_{t_{N}}, x_1)$ with **any** number $N\geq 1$ of intermediate points, which is Markovian, discrete reciprocal and shares the marginals $p(x_0)$ and $p(x_1)$. Even $N=1$ is enough in our framework, i.e., it is enough to find a joint distribution with three variables $\pi(x_0, x_t, x_1)$ and $t \in (0, 1)$, which is Markovian and discrete reciprocal to solve SB.
>
> We believe that our main theorem opens the door to a novel family of algorithms (including ASBM) for solving SB, based on finding a discrete probability distribution $\pi(x_0, x_{t_{1}}, \dots, x_{t_{N}}, x_1)$ that is Markovian and discrete reciprocal to solve the SB problem.
>
> We also believe that this theorem may facilitate the theoretical study of this problem, including the study on the convergence speed of D-IMF and IMF algorithms since it allows us to work with a more simple object — joint probability distribution instead of continuous stochastic processes. For instance, we show that it is possible to derive closed-form Markovian and reciprocal projections for a multivariate Gaussian case in our framework. In contrast, the continuous counterpart, form solution, is derived only for a 1-dimensional case [2].
>
> **(2) This model does not always yield state-of-the-art scores for syntactic datasets (see Table 1\&2).**
>
> Our algorithm does not beat all other competitors on all the setups. However, it provides the best results on most considered setups by both plan and target metrics (Tables 1 and 2).
>
> **(3) While I understand this work for some extent, I do not clearly understand the motivation why do we need a sampling-based approach when we have sampling-free method such as SF$^2$M [3] and LightSB [1] models.**
>
> In the limitation section, the LightSB [1] authors state that their algorithm is not designed for large-scale generative problems. Furthermore, they do not provide experiments in the image space, and they only consider image-to-image translation in the latent space of the autoencoder. Other experiments with real data include only single-cell data, which has lower dimensionality than images and does not lie on the low-dimensional manifold-like image distribution.
>
> The authors of SF2M [3] also consider only single-cell problems. Furthermore, while SF$^2$M [3] is indeed simulation-free, it theoretically guarantees convergence to SB only if samples from the ground-truth static SB solution $\pi^*(x_0, x_1)$ are used for training. However, if one has access to the $\pi^*(x_0, x_1)$, the problem itself is no longer unpaired domain translation since there is access to the ground-truth paired data. The authors of [3] address this problem by approximation of $\pi^*(x_0, x_1)$ by minibatch Optimal Transport techniques, which seems to provide a good approximation in the considered single-cell setups but is known to have a bias growing with the dimensionality of the considered space. For instance, no experiments show that this approximation works well in the case of high-dimensional unpaired image-to-image setups.
>
> Thus, while each of these simulation-free algorithms has its advantages and works well in the setups for which they were designed, there is still no simulation-free algorithm that can demonstrate good quality on the unpaired image-to-image translation without using any autoencoders.
>
> **(4) How do the authors suspect that the FID score is very low for ASBM? Following the arguments of [1], could the authors report CMMD scores or other metrics for results in Fig. 3?**
>
> Since it is hard to judge quality given only by the absolute values of FID, we compare our method with the DSBM and see that it provides better FID. As per your request, we measure the CMMD score and present the results below:
>
> |             | $\epsilon=1$ | $\epsilon=10$ |
> |-------------|--------------|---------------|
> | DSBM [2]    | 0.365        | 1.140         |
> | ASBM (ours) | 0.216        | 0.231         |
>
> We observe that CMMD scores align with FID scores presented in our work.
>
> **(5) Typo L213: Denoising Denoising GANs**
>
> Thank you for pointing out this. We will fix it in the final version.
>
> **Concluding remarks**.
> We would be grateful if you could let us know if the explanations we gave have been satisfactory in addressing your concerns about our work. If so, we kindly ask that you consider increasing your rating. We are also open to discussing any other questions you may have.
>
> References:
>
> [1] Alexander Korotin. Light schrödinger bridge. ICLR, 2024.
>
> [2] Shi et al. (2023) -- Diffusion Schrodinger Bridge Matching
>
> [3] Tong A. Y. et al. Simulation-Free Schrödinger Bridges via Score and Flow Matching AISTATS 2024.

---

> > ### Author Response · Authors · 2024-08-11
> >
> > Dear Reviewer LCNa,
> >
> > We thank you for your review and appreciate your time reviewing our paper.
> >
> > The end of the discussion period is close. We would be grateful if we could hear your feedback regarding our answers to the reviews. We are happy to address any remaining points during the remaining discussion period.
> >
> > Thanks in advance,
> >
> > Paper authors

---

> > > ### Comment · Reviewer_LCNa · 2024-08-11
> > >
> > > I would like to thank the authors for their clarifications and answers. After reading through other reviews and considering the pointed-out weaknesses, I decided to keep my original score.

---

### Official Review · Reviewer_RgPk · 2024-07-12

**Soundness:** 3
**Presentation:** 3
**Contribution:** 3
**Rating:** 6
**Confidence:** 3

**Summary:**

This paper proposes the Discrete Iterative Markovian Fitting (D-IMF) method. Specifically, this work introduces discretized reciprocal properties and Markovian processes, showing that the optimal plan matches the solution of the static Schrödinger Bridge (SB) problem. Additionally, this work demonstrate that the D-IMF procedure theoretically converges to the (static) SB solution. For the implementation of the Markovian projection, the authors employ a GAN structure (DD-GAN).

**Strengths:**

- The paper is theoretically fruitful. This work extends the reciprocal and Markovian properties into discrete sense, and also derive that the D-IMF method converges to static SB solution. This theoretical result is non-trivial and theoretically significant.
- Integrating these concepts into an adversarial framework is an innovative idea, adding value to the research.

**Weaknesses:**

- The algorithm seems to be ineffective. It requires two discriminator and two generator networks, which seems to require significant computational resource as well as significant computational time. It would be beneficial to compare the computational burden (both training time and gpu occupancy) with that of DSBM. Moreover, since it employs two adversarial networks, it would be hard to stabilize. It would be valuable to show its (empirical sense) stability through various ablation studies.

- The paper lacks of empirical studies. The practical experiment is only conducted on Male-Female data. Moreover, the paper lacks ablation studies. For example, the important hyperparameter such as the initial distribution setup (simple distribution produce vs minibatch matching), the number of function evaluations (NFE, DDGAN steps), and the number of phase (K) are not explored.

- The FID for Male-to-Female translation is reported, but not for the reverse direction. Since bidirectional learning is being conducted, it would be valuable to report results for both directions. Moreover, there is no metric provided to measure if the images are transformed into ones that are perceptually close. It would be useful to compare c-FID or Wasserstein distance with DSBM.

Minor comments:
There is a typo in Line 224: the term $q(x_{t_n}|x_{t_{n-1}})$.

**Questions:**

- Can the authors provide a comparison of the computational burden between your method and DSBM, particularly in terms of time and resource usage?
- Is the time discretization handled as like DDPM style or Uniformly?
- Did you use any pretrained model (e.g. pretrained DSBM) on I2I task?

**Limitations:**

The limitation is adequately discussed in the paper.

---

> ### Author Rebuttal · Authors · 2024-08-07
>
> Dear Reviewer RgPk, thank you for your comments. Here are the answers to your questions and comments.
>
> **(1) Computational burden and stability study.**
>
> As requested, we provide a study on the effectiveness of ASBM and compare it with the DSBM.
>
> **Number of parameters.** Our ASBM generator has about 42 million parameters, while each discriminator has 27 million. The generator of DSBM has 38 million parameters. Thus, generators have similar capacities.
>
> **Training time.** We mentioned (lines 714-716) that for Celeba setups, our ASBM uses 7 days of training on a A100 GPU, while the DSBM uses 5 days of training. The time is comparable since both methods use the same number of IMF and D-IMF iterations.
>
> Please note that we provide an analysis of ASBM (ours) and DSBM algorithms in Appendix E, where we show that additional iterations only slightly change the results.
>
> **Inference time.** We measure the inference time of both ASBM (ours) and DSBM algorithms on the same hardware with one A100 GPU. We measure the mean and standard deviation over 16 runs. For a batch of 100 images, ASBM requires about 842 ms (std 97 ms), while DSBM requires 33870 ms (std 47 ms). **Thus, our algorithm provides up to 40x speedup compared to DSBM.**
>
> **GPU memory consumption.** On the Celeba setup, our algorithm requires about 25 GB of GPU for training with a batch size of 32 and about 12 GB on the inference stage to translate a batch of 100 images. Meanwhile, DSBM requires 34 GB of GPU for training with batch size 32 and about 26.7 GB on the inference stage for a batch of 100. Thus, our algorithm utilizes significantly less GPU memory.
>
> **Stability of training.**
> To show that our procedure is stable, we provide the plot of FID during the training of all D-IMF iterations in Figure 3 of the **attached pdf**. On the plot, we provide FID vs the number of generator gradients updated during the training. We highlight the starts of each D-IMF iteration by the vertical lines. We observe stable decreases of FID.
>
> **(2) Lack of empirical and ablation studies.**
>
> **The number of phases K.** We provide an analysis of D-IMF/IMF iterations (K) in Appendix E.1. of our work.
>
> **Independent vs minibatch coupling.**
> As per request, we provide a study on running ASBM with different coupling. We provide the additional results for the Colored MNIST setup with $\epsilon=10$ (which we discuss in Appendix C.3 of our paper). In Figure 1 of **attached pdf**, we provide qualitative generation results for our ASBM model with independent and minibatch couplings after 3 D-IMF iterations, which we used in our work for this setup. We also provide FID values in Table 1 of **attached pdf**. We see comparable visual quality as well as similarity for both results; the FID metric is slightly better for minibatch case.
>
> **Study on different NFE.**
> As requested, we present the study's results on different NFE of our ASBM model. In our study, we use our ASBM model trained with $4$ steps for Celeba male to female with $\epsilon=1.0$. We test this model with different numbers of NFE $\in [1, 2, 3, 4, 8, 16, 32]$. In all the cases, we use a uniform schedule. (See answer 4 for details). We present qualitative results of generation for several inputs in Figure 2 of the **attached pdf**. We also present the FID values computed on the test set and average $L_2$ cost $c(x_0,x_1) = \frac{||x_0-x_1||^2}{2}$ between the input image $x_0$ and generated image $x_1$ to assess similarity in Table 2 of the **attached pdf**. Interestingly, our model with $\text{NFE}=3$ provides almost the same FID value as for $\text{NFE}=4$, which we use at the training, but significantly lower $L_2$ cost.
>
> **Training with different NFE.**
> Since the time of the rebuttle and our resources are limited we train ASBM with $NFE=2$ and $NFE=8$ with $K=2$ D-IMF iterations and $\epsilon=1.0$ using the same procedure as before. We present samples in the Figure 5 together with ASBM model with $NFE=4$ after $K=2$ D-IMF iteration trained before. Male-to-female metrics:
>
> ||NFE=2|NFE=4|NFE=8|
> |-|-|-|-|
> |FID|11.829|16.08|29.58|
> |LPIPS|0.214|0.253|0.241|
>
> We observe better results for lower NFE, possibly because the model needs to learn fewer transitional distributions. We will add all the results to the final version.
>
> **(3) The FID of the reverse direction.**
>
> Thank you for this suggestion. The values are given below.
>
> ||$\epsilon=1$|$\epsilon=10$|
> |-|-|-|
> |DSBM [2]|24.06|92.15|
> |ASBM (ours)|16.86 |17.44|
>
> Thus, the FID values for the reverse direction show the same behavior as for the forward direction.
>
> **(4) Measurement of perceptual quality**
>
> As requested, we measure perceptual similarity between input and translated images using standard LPIPS metric [1] (as used in DSBM). For our Celeba male-to-female setup, we measure LPIPS between input and translated images from the test. We present the results below (the lower the better):
>
> || $\epsilon=1$ | $\epsilon=10$ |
> |-|-|-|
> |DSBM [2]|0.246| 0.386|
> |ASBM (ours)|0.242|0.294|
>
> We observe almost the same similarity in the case of $\epsilon=1$ and better similarity for ASBM (ours) in the case of $\epsilon=10$. We will add these results in the final version.
>
> **(5) Time discretization.**
>
> We use uniform time discretization in all our experiments, i.e., for the number of inner times points $N$, we use time discretization $t_{n} = \frac{n}{N+1}$ for $n \in [0, N+1]$.
>
> **(6) Did you use any pretrained model?**
>
> No, we do not use any pretrained models in our experiments.
>
> **Concluding remarks**.
> We would be grateful if you could let us know if the explanations we gave have been satisfactory in addressing your concerns and questions about our work. If so, we kindly ask that you consider increasing your rating. We are also open to discussing any other questions you may have.
>
> References:
>
> [1] Zhang R. The unreasonable effectiveness of deep features as a perceptual metric. CVPR, 2018.
>
> [2] Shi Y. Diffusion Schrödinger bridge matching. NeurIPS, 2024.

---

> > ### Comment · Reviewer_RgPk · 2024-08-11
> > **Thank you for the response**
> >
> > I appreciate authors for clarification and for the additional ablation studies. I would like to raise my score to 6.

---

### Official Review · Reviewer_xGf5 · 2024-07-12

**Soundness:** 3
**Presentation:** 3
**Contribution:** 3
**Rating:** 7
**Confidence:** 4

**Summary:**

The paper extends the recently proposed Schrödinger Bridge method, DSBM or IMF, to a discrete-time setup. This extension is non-trivial, requiring appropriate notation for discrete reciprocal and Markovian projections. By leveraging the structure of Brownian bridges used in the original IMF, the paper achieves efficient sampling and inference processes. The discrete-time formulation also establishes a natural connection to other generative model classes, such as GANs. Experiments were conducted on low-dimensional Gaussians and unsupervised image-to-image translation.

**Strengths:**

- The discrete-time construction of IMF is an elegant piece derived from the DSBM, which heavily relies on Brownian bridges. The analytic solutions for Gaussians may also be of interest to readers from other domains.

- The connection to DD-GANs is both straightforward and clever. It's exciting to see how different classes of generative models are starting to overlap and combine, leveraging the strengths of each approach.

**Weaknesses:**

- I enjoyed reading the majority of the paper—except, perhaps oddly, the paper title. Over 85% of the main technical contributions (Sec 3) stem from the construction of the discrete IMF formulation. The concept of "adversarial" appears only as one (of many plausible) implementation of Eq (15). I feel like the "discrete" aspect isn't highlighted enough, or at all, in the title. This is, of course, just a personal comment rather than a criticism.

- In L220, the non-saturating GAN loss should be provided in the main paper rather than being postponed to the Appendix.

**Questions:**

- The notions of $T$ and $M$ are slightly confusing. For example, In L73 and L76, both represent Markovian processes. Is there a typo?

**Limitations:**

Limitations were addressed in Appendix A.

---

> ### Author Rebuttal · Authors · 2024-08-07
>
> Dear Reviewer xGf5, thank you for your comments. Here are the answers to your questions and comments.
>
> **(1) I enjoyed reading the majority of the paper—except, perhaps oddly, the paper title. Over 85\% of the main technical contributions (Sec 3) stem from the construction of the discrete IMF formulation. The concept of "adversarial" appears only as one (of many plausible) implementation of Eq (15). I feel like the "discrete" aspect isn't highlighted enough, or at all, in the title. This is, of course, just a personal comment rather than a criticism.**
>
> The title of the work which introduces DSBM algorithm is "Diffusion Schrödinger Bridge Matching".  Our work is based on it but, at the same time, proposes to use adversarial training as a current main competitor of diffusion models in the field of generative models. To highlight both the relation with the prior work and the usage of adversarial training in our approach, we decide to use the title "Adversarial Schrödinger Bridge Matching".
>
> **(2) In L220, the non-saturating GAN loss should be provided in the main paper rather than being postponed to the Appendix.**
>
> Thank you for this suggestion. We initially placed the non-saturating GAN loss in the Appendix to keep the main text focused and avoid introducing technical details too early. Furthermore, the non-saturating GAN loss is rather standard approach in GANs, so we did not pay much attention to it. However, we agree that it would be beneficial to mention the non-saturating GAN loss in the main paper for clarity. We will move it to the main text in the final version and refer readers to the Appendix for a more detailed description of our adversarial procedure.
>
> **(3) The notions of T and M are slightly confusing. For example, In L73 and L76, both represent Markovian processes. Is there a typo?**
>
> In this context, the process $T$ and $M$ are the same Markovian process. We will replace $T$ with $M$ in the final version. Thank you for mentioning it.
>
> **Concluding remarks**.
> We would be grateful if you could let us know if the explanations we gave have been satisfactory in addressing your concerns and questions about our work.  We are also open to discussing any other questions you may have.

---

> > ### Author Response · Authors · 2024-08-11
> >
> > Dear Reviewer xGf5,
> >
> > We thank you for your review and appreciate your time reviewing our paper.
> >
> > The end of the discussion period is close. We would be grateful if we could hear your feedback regarding our answers to the reviews. We are happy to address any remaining points during the remaining discussion period.
> >
> > Thanks in advance,
> >
> > Paper authors

---

> > ### Comment · Reviewer_xGf5 · 2024-08-11
> >
> > I thank the authors for the reply. I've decided to keep my score.

---

### Official Review · Reviewer_xtGA · 2024-07-30

**Soundness:** 3
**Presentation:** 3
**Contribution:** 3
**Rating:** 6
**Confidence:** 4

**Summary:**

In this paper, the authors study the Schrodinger Bridge problem and propose an improvement over the Diffusion Schrodinger Bridge Matching (DSBM) methodology [1, 2]. In particular, they propose a discrete counterpart formulation of DSBM. Instead of relying on stochastic processes, they rely on a discrete-time version of the Markovian and reciprocal projections which are the key components of DSBM. By doing so, they have a better control on the number of steps needed in DSBM. They next leverage advances in the field of diffusion models to reduce the number of stepsizes needed in order to solve the Schrodinger Bridge. In particular, they focus on larger jumps using the approach of Denoising Diffusion GAN (DD-GAN), introduced in [3]. The method is illustrated on translation tasks on the Celeba dataset.

[1] Shi et al. (2023) -- Diffusion Schrodinger Bridge Matching

[2] Peluchetti (2023) -- Diffusion Bridge Mixture Transports, Schrödinger Bridge Problems and Generative Modeling

[2] Xiao et al. (2021) -- Tackling the Generative Learning Trilemma with Denoising Diffusion GANs

**Strengths:**

* The paper is overall clearly written and easy to follow. It relies on the notions of Markovian and reciprocal projections that were already introduced in [1]. All the required notions are properly introduced.

* I enjoy reading the full description of the DSBM algorithm in discrete time. I agree that this should allow for 1) easier spreading of DSBM-related methodology 2) more flexible algorithm (like the introduction of GAN jumps as proposed).

* The experimental results seem to compare favorably compared to DSBM (more on that later).

[1] Shi et al. (2023) -- Diffusion Schrodinger Bridge Matching

**Weaknesses:**

* Part of the theory was already done in [1]. Especially the discrete Markovian projection was already identified as the minimizer of the Kullback-Leibler divergence in [1, Appendix E]. I think the claims of novelty of the paper regarding the introduction of the discrete scheme should be tamed down.

* I am not clear regarding what is the purpose of the Gaussian study. I understand that the authors use it in their experimental section (Section 4.1) to study the convergence of their algorithm but I think Section 3.4 is distracting the reader from what I consider to be the main points of the paper which is 1) full discrete study of DSBM 2) GAN jumps. Section 3.4 could easily be considered in the Appendix which would leave more room for additional experiments.

* The claim of exponential convergence should be tamed down. Reading the paper it seems like this is proven in the current paper (I am talking especially about the bold "exponential rate of convergence" in l.294 which can be misleading). To my understanding this is only experimentally shown.

* There is something quite striking in Figure 3. (for example see the last row of (d) and (f)). While the quality of ASBM is better (by the way, one of the reason the FID score is so high for DSBM must be because of the remaining noise in the output samples, an easy way to remove this noise would be to consider a deterministic last step as a postprocessing as is routinely done in diffusion models, see [2] for instance), it seems that the diversity is much lower than DSBM. I would like to see a proper investigation of this phenomenon which is also reported in the adversarial distillation of diffusion models. According to [3], there is an incompressible trade-off between high-NFE/good diversity and low-NFE/low-diversity.

[1] Shi et al. (2023) -- Diffusion Schrodinger Bridge Matching

[2] Jolicoeur et al. (2020) -- Adversarial score matching and improved sampling for image generation

[3] Dieleman (2024) -- The paradox of diffusion distillation

**Questions:**

* I understand that the main goal of the author is the speed-up of DSBM-like algorithms. However, why choose DD-GAN? This doesn't seem to me like the obvious candidate for speeding up DSBM. Other techniques like distillation (adversarial distillation if one really wants to use GANs [1, 2]) or consistency models seem like potential better candidate. If the finetuning strategy is what the authors want to avoid, consistency models can be trained from scratch [3, 4].

* Some typos:

* l.280 "Schrodigner" --> "Schrodinger"

[1] Xu et al. (2023) --UFOGen: You Forward Once Large Scale Text-to-Image Generation via Diffusion GANs

[2] Sauer et al. (2023) -- Adversarial Diffusion Distillation

[3] Song et al. (2023) -- Consistency Models

[4] Song et al. (2023) -- Improved Techniques for Training Consistency Models

**Limitations:**

The limitations are discussed in the Appendix A. However, the authors are quick to dismiss these limitations. For example, regarding the adversarial training "While we mention this aspect, we do not treat it to a be a serious limitation." I think there are serious problems regarding adversarial training regarding the training (suffer from training instabilities) and the results themselves (less diversity). These issues are usually reported in the field of adversarial distillation of diffusion models, see [1] for instance. I think this is doing a disservice to the paper and the community by trying to minimise them instead of exposing them for what they are.

[1] Dieleman (2024) -- The paradox of diffusion distillation

---

> ### Author Rebuttal · Authors · 2024-08-05
>
> Dear Reviewer xtGA, thank you for your comments. Here are the answers to your questions and comments.
>
> **(1) Part of the theory was already done in [1]. ...**
>
> Indeed, the authors of [1] analyze the discrete-time Markovian projection and show that it converges to continuous-time projection in certain limiting cases. We cite this result in line 260. We will add additional citations and discussion in Section 3.3 after defining Markovian projection to highlight some aspects that have already been introduced in [1].
>
> Please note that our Proposition 3.5 on KL-minimization extends the results from [1, Appendix E]. The authors of [1] considered the Markovian projection for a mixture of **Markovian** bridges. In turn, in our Proposition 3.5 we consider arbitrary distribution $\pi(x_0, x_{t_{1}}, \dots, x_{t_{N}}, x_1)$ without any assumptions on its conditional distribution $\pi(x_{t_{1}}, \dots, x_{t_{N}}|x_0, x_1)$. It makes the proof technically more complex but provides a more general result.
>
> **(2) ... the purpose of the Gaussian study. ...**
>
> There are two purposes: to experimentally study the rate of convergence and to demonstrate that a closed-form solution for the multivariate Gaussian case can be derived within the discrete-IMF framework. In the final version, we will shorten Section 3.4 by moving detailed technical definitions (such as matrices $U$ and $K$) to the Appendix but retain the key points in the main text.
>
> **(3) The claim of exponential convergence should be tamed down. ...**
>
> We agree that the phrasing in line 294 could be misleading. This observation is based solely on our experimental results, and the whole discussion on convergence rate is placed inside the experimental section (Section 4), while all of our theory is placed in Section 3. We also directly mentioned that there is no theoretical proof in the limitation section (lines 465-467). To avoid possible misunderstanding, we will revise the bolded text.
>
> **(4) ... remove this noise ... consider a deterministic last step  as a postprocessing ...**
>
> We use the original published code from the DSBM authors [1] to ensure a fair comparison. In this code, noise is not added at the last sampling step. As you requested, we additionally used the drift function for the last time-moment $t=0.99$ to add an extra denoising step at the final step of the sampling process. It indeed removes the remaining noise, (see the samples in Figure 4 of the **attached pdf**). For the Celeba setup with $\epsilon=1.0$, we observe the improvement of the FID metric from $37.8$ to $29.7$.   We will add this additional result in the experimental section.
>
> **(5) ... it seems that the diversity is much lower than DSBM. I would like to see a proper investigation of this phenomenon ...**
>
> As requested, we provide a study on the diversity of images produced. We measure the diversity of images produced by DSBM and ASBM on the Celeba setup by utilizing the LPIPS diversity as was done in [3, Table 1]. We take a subset of $500$ images from the test part of the Celeba dataset. We sample a batch of generated images of size $16$ for each input image, compute the average distance between all possible pairs of these images, and average these values over all inputs. We present the results below for DSBM and ASBM and different values of the coefficient $\epsilon=1$ and $\epsilon=10$.
>
> || $\epsilon=1$ | $\epsilon=10$ |
> |-------------|--------------|---------------|
> |DSBM [1]| 0.1047| 0.1909|
> |ASBM (ours) | 0.0933| 0.1878|
>
> Indeed, the LPIPS diversity is lower for ASBM. However, without a known ground truth for diversity, it is difficult to determine definitively whether ASBM provides insufficient diversity or if DSBM produces too much. We will include these findings in the final version of the paper and discuss this trade-off.
>
> **(6) .... However, why choose DD-GAN? ...**
>
> The mentioned adversarial distillation techniques [1,2] are based on tractable forward process in diffusion models for simulation-free sampling, which is not feasible for the unpaired Schrödinger Bridge (SB). Additionally, these methods require an already trained model for distillation, whereas our approach does not need a pretrained diffusion model.
>
> Regarding consistency models, it is unclear how to use them in an unpaired image-to-image setup. Works [3, 4] do not address this. The work [3] discusses generative models and zero-shot translation with known degradations, applicable only to specific tasks like colorization. Similarly, [4] focuses on generative modeling. While consistency models can potentially accelerate arbitrary ODEs, the Schrödinger Bridge (SB) problem's solution is an SDE, which makes it challenging to use consistency models.
>
> **(7) ...the authors are quick to dismiss these limitations. ... (Appendix A)**
>
> We understand that adversarial training is a limitation. We intended to say that we use the DD-GAN approach, designed to address this limitation to some extent (mode coverage [2, Section 5.3], training stability [2, Appendix D]).
>
> Furthermore, to show that our DDGAN-based ASBM procedure is stable, we additionally provide the plot of FID during the training of all D-IMF iterations in Figure 3 of the **attached pdf**. We provide FID vs the number of generator gradients updated during the training. We highlight the starts of each D-IMF iteration by the vertical lines. We observe the stable decrease of FID over the generator gradient update steps.
>
> We will remove the phrase "we do not treat it to a be a serious limitation" in the final version.
>
> **Concluding remarks**.
> We would be grateful if you could let us know if the explanations we gave have been satisfactory in addressing your concerns about our work. If so, we kindly ask that you consider increasing your rating. We are also open to discussing any other questions you may have.
>
> **References** are the same as you used in the corresponding weaknesses and questions.

---

> > ### Author Response · Authors · 2024-08-11
> >
> > Dear Reviewer xtGA,
> >
> > We thank you for your review and appreciate your time reviewing our paper.
> >
> > The end of the discussion period is close. We would be grateful if we could hear your feedback regarding our answers to the reviews. We are happy to address any remaining points during the remaining discussion period.
> >
> > Thanks in advance,
> >
> > Paper authors

---

> > > ### Comment · Reviewer_xtGA · 2024-08-12
> > >
> > > I would like to thank the authors for their thorough rebuttal. While the authors have answered to my main questions I would like to keep my score to 6. In case the paper is accepted I think the discussion on diversity of the samples should be included. This is actually a really good starting point for a wider discussion on the use of GANs methods within diffusion models.

---

### Author Rebuttal · Authors · 2024-08-07

Dear reviewers, thank you all for taking the time to review our paper and for your thoughtful feedback. We are delighted that you found our work to be well-written and that it effectively introduces key concepts (xtga, LCNa). It is encouraging to hear that you see it as theoretically fruitful (RgPk), that it broadens our understanding of IMF (LCNa), and that it introduces a straightforward and clever connection between IMF and DD-GANs (xGf5). We are also pleased that you believe our work allows for easier spreading of DSBM-related methodology and developing more flexible algorithms (xtga).

Please find the answers to your questions below. **Please note that we have added tables and figures in the attached pdf to support our responses to the reviewers xtGA and RgPk.**

---

### Decision · Program_Chairs · 2024-09-25

**Decision:**

Accept (poster)

**Comment:**

This paper proposes a novel Discrete-time Iterative Markovian Fitting (D-IMF) procedure for solving the Schrödinger Bridge (SB) problem, aiming to improve the efficiency of the existing Iterative Markovian Fitting (IMF) method. By leveraging the Denoising Diffusion GAN (DD-GAN) framework, the authors achieve comparable results to IMF with fewer steps.

Reviewers generally agree that the paper presents an interesting and principled approach, and it is well-written. However, to maximize the paper's impact, the authors should carefully address the remaining concerns raised by reviewers in the final version. The experimental section should be strengthened to include detailed comparisons with DSBM, ablation studies, and experiments beyond the Celeba dataset. Additionally, the computational complexity of the method, particularly the use of two discriminator and two generator networks, should be discussed more thoroughly, as this might be practically costly and potentially lead to instability during adversarial training.